# A Deep Variational Approach to Clustering Survival Data

**Laura Manduchi,**[†1] **Ričards Marcinkevičs,**[†1] **Michela C. Massi,**[2,3] **Thomas Weikert,**[4]
**Alexander Sauter,**[4] **Verena Gotta,**[5] **Timothy Müller,**[6] **Flavio Vasella,**[6] **Marian C. Neidert,**[7]
**Marc Pfister,**[5] **Bram Stieltjes**[4] **& Julia E. Vogt**[1]

[1]ETH Zürich; [2]Politecnico di Milano; [3]CADS, Human Technopole; [4]University Hospital Basel;
[5]University Children's Hospital Basel; [6]University of Zürich; [7]St. Gallen Cantonal Hospital

## Abstract

In this work, we study the problem of clustering survival data — a challenging and so far under-explored task. We introduce a novel semi-supervised probabilistic approach to cluster survival data by leveraging recent advances in stochastic gradient variational inference. In contrast to previous work, our proposed method employs a deep generative model to uncover the underlying distribution of *both* the explanatory variables and censored survival times. We compare our model to the related work on clustering and mixture models for survival data in comprehensive experiments on a wide range of synthetic, semi-synthetic, and real-world datasets, including medical imaging data. Our method performs better at identifying clusters and is competitive at predicting survival times. Relying on novel generative assumptions, the proposed model offers a holistic perspective on clustering survival data and holds a promise of discovering subpopulations whose survival is regulated by different generative mechanisms.

## 1 Introduction

Survival analysis (Rodríguez, 2007; D. G. Altman, 2020) has been extensively used in a variety of medical applications to infer a relationship between explanatory variables and a potentially *censored* survival outcome. The latter indicates the time to a certain event, such as death or cancer recurrence, and is *censored* when its value is only partially known, *e.g.* due to withdrawal from the study (see Appendix A). Classical approaches include the Cox proportional hazards (PH; Cox (1972)) and accelerated failure time (AFT) models (Buckley & James, 1979). Recently, many machine learning techniques have been proposed to learn nonlinear relationships from unstructured data (Faraggi & Simon, 1995; Ranganath et al., 2016; Katzman et al., 2018; Kvamme et al., 2019).

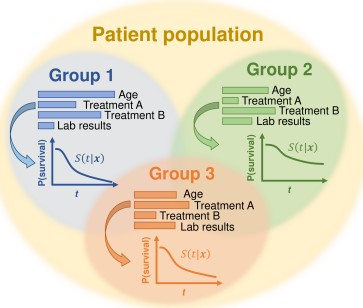

Figure 1: Survival clustering.

Clustering, on the other hand, serves as a valuable tool in data-driven discovery and subtyping of diseases. Yet a fully unsupervised clustering algorithm does not use, by definition, the survival outcomes to identify clusters. Therefore, there is no guarantee that the discovered subgroups are correlated with patient survival (Bair & Tibshirani, 2004). For this reason, we focus on a semi-supervised learning approach to cluster survival data that jointly considers explanatory variables and censored outcome as indicators for a patient's state. This problem is particularly relevant for precision medicine (Collins & Varmus, 2015). The identification of such patient subpopulations could, for example, facilitate a better understanding of a disease and a more personalised disease management (Fenstermacher et al., 2011). Figure 1 schematically depicts this clustering problem: here, the overall patient population consists of three groups characterised by different *associations* between the covariates and survival, resulting in disparate clinical conditions. The survival distributions do not need to differ between clusters: compare groups 1 (■) and 3 (■).

---

[†]Equal contribution. Correspondence to {laura.manduchi,ricardsm}@inf.ethz.ch

Table 1: Comparison of the proposed model to the related approaches: semi-supervised clustering (SSC), profile regression (PR) for survival data, survival cluster analysis (SCA), and deep survival machines (DSM). Here, $t$ denotes survival time, $x$ denotes explanatory variables, $z$ corresponds to latent representations, $K$ stands for the number of clusters, and $\mathcal{L}(\cdot, \cdot)$ is the likelihood function.

| | SSC | PR | SCA | DSM | VaDeSC |
|---|---|---|---|---|---|
| Predicts $t$? | ✗ | ✓ | ✓ | ✓ | ✓ |
| Learns $z$? | ✗ | ✗ | ✓ | ✓ | ✓ |
| Maximises $\mathcal{L}(x, t)$? | ✗ | ✓ | ✗ | ✗ | ✓ |
| Scalable? | ✗ | ✗ | ✓ | ✓ | ✓ |
| Does not require $K$? | ✗ | ✓ | ✓ | ✗ | ✗ |

Clustering of survival data, however, remains an under-explored problem. Only few methods have been proposed in this context and they either have limited scalability in high-dimensional, unstructured data (Liverani et al., 2020), or they focus on the discovery of purely *outcome-driven* clusters (Chapfuwa et al., 2020; Nagpal et al., 2021a), that is clusters characterised entirely by survival time. The latter might fail in applications where the survival distribution alone is not sufficiently informative to stratify the population (see Figure 1). For instance, groups of patients characterised by similar survival outcomes might respond very differently to the same treatment (Tanniou et al., 2016).

To address the issues above, we present a novel method for clustering survival data — *variational deep survival clustering* (VaDeSC) that discovers groups of patients characterised by different generative mechanisms of survival outcome. It extends previous variational approaches for unsupervised deep clustering (Dilokthanakul et al., 2016; Jiang et al., 2017) by incorporating cluster-specific survival models in the generative process. Instead of only focusing on survival, our approach models the heterogeneity in the *relationships* between the covariates and survival outcome.

**Our main contributions** are as follows: (*i*) We propose a novel, deep probabilistic approach to survival cluster analysis that jointly models the distribution of explanatory variables and censored survival outcomes. (*ii*) We comprehensively compare the clustering and time-to-event prediction performance of VaDeSC to the related work on clustering and mixture models for survival data on synthetic and real-world datasets. In particular, we show that VaDeSC outperforms baseline methods in terms of identifying clusters and is comparable in terms of time-to-event predictions. (*iii*) We apply our model to computed tomography imaging data acquired from non-small cell lung cancer patients and assess obtained clustering qualitatively. We demonstrate that VaDeSC discovers clusters associated with well-known patient characteristics, in agreement with previous medical findings.

## 2 RELATED WORK

Clustering of survival data has been first explored by Bair & Tibshirani (2004) (semi-supervised clustering; SSC). The authors propose pre-selecting variables based on univariate Cox regression hazard scores and subsequently performing $k$-means clustering on the subset of features to discover patient subpopulations. More recently, Ahlqvist et al. (2018) use Cox regression to explore differences across subgroups of diabetic patients discovered by $k$-means and hierarchical clustering. In the spirit of the early work by Farewell (1982) on mixtures of Cox regression models, Mouli et al. (2018) propose a deep clustering approach to differentiate between long- and short-term survivors based on a modified Kuiper statistic in the absence of end-of-life signals. Xia et al. (2019) adopt a multitask learning approach for the outcome-driven clustering of acute coronary syndrome patients. Chapfuwa et al. (2020) propose a survival cluster analysis (SCA) based on a truncated Dirichlet process and neural networks for the encoder and time-to-event prediction model. Somewhat similar techniques have been explored by Nagpal et al. (2021a) who introduce finite Weibull mixtures, named deep survival machines (DSM). DSM fits a mixture of survival regression models on the representations learnt by an encoder neural network. From the modelling perspective, the above approaches focus on *outcome-driven* clustering, i.e. they recover clusters entirely characterised by different survival distributions. On the contrary, we aim to model cluster-specific *associations* between covariates and survival times to discover clusters characterised not only by disparate risk but also by different survival generative mechanisms (see Figure 1). In the concurrent work, Nagpal et al. (2021b) introduce deep Cox mixtures (DCM) jointly fitting a VAE and a mixture of Cox regressions. DCM does not specify a generative model and its loss is derived empirically by combining the VAE loss with the

likelihood of survival times. On the contrary, our method is probabilistic and has an interpretable generative process from which an ELBO of the joint likelihood can be derived.

The approach by Liverani et al. (2020) is, on the other hand, the most closely related to ours. The authors propose a clustering method for collinear survival data based on the profile regression (PR; Molitor et al. (2010)). In particular, they introduce a Dirichlet process mixture model with cluster-specific parameters for the Weibull distribution. However, their method is unable to tackle high-dimensional unstructured data, since none of its components are parameterised by neural networks. This prevents its usage on real-world complex datasets, such as medical imaging (Haarburger et al., 2019; Bello et al., 2019). Table 1 compares our and related methods w.r.t. a range of properties. For an overview of other lines of work and a detailed comparison see Appendices B and C.

## 3 METHOD

We present VaDeSC — a novel variational deep survival clustering model. Figure 2 provides a summary of our approach: the input vector $x$ is mapped to a latent representation $z$ using a VAE with a Gaussian mixture prior. The survival density function is given by a mixture of Weibull distributions with cluster-specific parameters $\beta$. The parameters of the Gaussian mixture and Weibull distributions are then optimised jointly using both the explanatory input variables and survival outcomes.

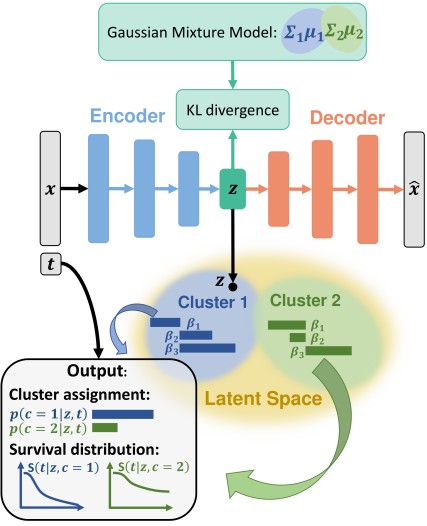

Figure 2: Summary of the VaDeSC.

**Preliminaries** We consider the following setting: let $\mathcal{D} = \{(x_i, \delta_i, t_i)\}_{i=1}^{N}$ be a dataset of $N$ three-tuples, one for each patient. Herein, $x_i$ denotes the explanatory variables, or features. $\delta_i$ is the censoring indicator: $\delta_i = 0$ if the survival time of the $i$-th patient was censored, and $\delta_i = 1$ otherwise. Finally, $t_i$ is the potentially censored survival time. A maximum likelihood approach to survival analysis seeks to model the survival distribution $S(t|x) = P(T > t|x)$ (Cox, 1972). Two challenges of survival analysis are (*i*) the censoring of survival times and (*ii*) a complex nonlinear relationship between $x$ and $t$. When clustering survival data, we additionally consider a latent cluster assignment variable $c_i \in \{1, ..., K\}$ unobserved at training time. Here, $K$ is the total number of clusters. The problem then is twofold: (*i*) to infer unobserved cluster assignments and (*ii*) model the survival distribution given $x_i$ and $c_i$.

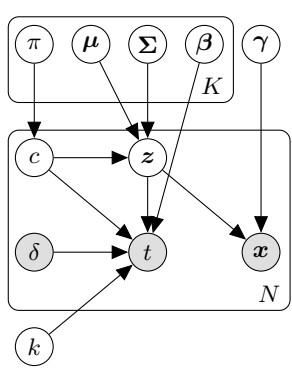

Figure 3: Generative model.

**Generative Model** Following the problem definition above, we assume that data are generated from a random process consisting of the following steps (see Figure 3). First, a cluster assignment $c \in \{1, ..., K\}$ is sampled from a categorical distribution: $c \sim p(c; \boldsymbol{\pi}) = \pi_c$. Then, a continuous latent embedding, $z \in \mathbb{R}^J$, is sampled from a Gaussian distribution, whose mean and variance depend on the sampled cluster $c$: $z \sim p(z|c; \{\boldsymbol{\mu}_1, ..., \boldsymbol{\mu}_K\}, \{\boldsymbol{\Sigma}_1, ..., \boldsymbol{\Sigma}_K\}) = \mathcal{N}(z; \boldsymbol{\mu}_c, \boldsymbol{\Sigma}_c)$. The explanatory variables $x$ are generated from a distribution conditioned on $z$: $x \sim p(x|z; \boldsymbol{\gamma})$, where $p(x|z; \boldsymbol{\gamma}) = \text{Bernoulli}(x; \boldsymbol{\mu}_{\boldsymbol{\gamma}})$ for binary-valued features and $\mathcal{N}(x; \boldsymbol{\mu}_{\boldsymbol{\gamma}}, \text{diag}(\boldsymbol{\sigma}_{\boldsymbol{\gamma}}^2))$ for real-valued features. Herein, $\boldsymbol{\mu}_{\boldsymbol{\gamma}}$ and $\boldsymbol{\sigma}_{\boldsymbol{\gamma}}^2$ are produced by $f(z; \boldsymbol{\gamma})$ – a decoder neural network parameterised by $\boldsymbol{\gamma}$. Finally, the survival time $t$ depends on the cluster assignment $c$, latent vector $z$, and censoring indicator $\delta$, *i.e.* $t \sim p(t|z, c)$. Similarly to conventional survival analysis, we assume *non-informative* censoring (Rodríguez, 2007).

**Survival Model** Above, $p(t|z, c)$ refers to the cluster-specific survival model. We follow an approach similar to Ranganath et al. (2016) and Liverani et al. (2020); in particular, we as-

sume that given $\boldsymbol{z}$ and $c$, the uncensored survival time follows the Weibull distribution given by Weibull $\left(\text{softplus}\left(\boldsymbol{z}^\top \boldsymbol{\beta}_c\right), k\right)$, where $\text{softplus}(x) = \log\left(1 + \exp(x)\right)$; $k$ is the shape parameter; and $\boldsymbol{\beta}_c$ are cluster-specific survival parameters. Note that we omitted the bias term $\beta_{c,0}$ for the sake of brevity. Observe that $\text{softplus}\left(\boldsymbol{z}^\top \boldsymbol{\beta}_c\right)$ corresponds to the scale parameter of the Weibull distribution. We assume that the shape parameter $k$ is global; however, an adaptation to cluster-specific parameters, as proposed by Liverani et al. (2020), is straightforward. The Weibull distribution with scale $\lambda$ and shape $k$ has the probability density function given by $f(x;\ \lambda, k) = \frac{k}{\lambda}\left(\frac{x}{\lambda}\right)^{k-1}\exp\left(-\left(\frac{x}{\lambda}\right)^k\right)$, for $x \geq 0$. Consequently, adjusting for right-censoring yields the following distribution:

$$
\begin{aligned}
p(t|\boldsymbol{z}, c;\ \boldsymbol{\beta}, k) &= f(t;\ \lambda_c^{\boldsymbol{z}}, k)^\delta S(t|\boldsymbol{z}, c)^{1-\delta} \\
&= \left[\frac{k}{\lambda_c^{\boldsymbol{z}}}\left(\frac{t}{\lambda_c^{\boldsymbol{z}}}\right)^{k-1}\exp\left(-\left(\frac{t}{\lambda_c^{\boldsymbol{z}}}\right)^k\right)\right]^\delta \left[\exp\left(-\left(\frac{t}{\lambda_c^{\boldsymbol{z}}}\right)^k\right)\right]^{1-\delta},
\end{aligned}
\tag{1}
$$

where $\boldsymbol{\beta} = \{\boldsymbol{\beta}_1, \ldots, \boldsymbol{\beta}_K\}$; $\lambda_c^{\boldsymbol{z}} = \text{softplus}\left(\boldsymbol{z}^\top \boldsymbol{\beta}_c\right)$; and $S(t|\boldsymbol{z}, c) = \int_{t=t}^\infty f(t;\ \lambda_c^{\boldsymbol{z}}, k)$ is the survival function. Henceforth, we will use $p(t|\boldsymbol{z}, c)$ as a shorthand notation for $p(t|\boldsymbol{z}, c;\ \boldsymbol{\beta}, k)$. In this paper, we only consider right-censoring; however, the proposed model can be extended to tackle other forms of censoring.

**Joint Probability Distribution**  Assuming the generative process described above, the joint probability of $\boldsymbol{x}$ and $t$ can be written as $p(\boldsymbol{x}, t) = \int_{\boldsymbol{z}} \sum_{c=1}^K p(\boldsymbol{x}, t, \boldsymbol{z}, c) = \int_{\boldsymbol{z}} \sum_{c=1}^K p(\boldsymbol{x}|t, \boldsymbol{z}, c)p(t, \boldsymbol{z}, c)$. It is important to note that $\boldsymbol{x}$ and $t$ are independent given $\boldsymbol{z}$, so are $\boldsymbol{x}$ and $c$. Hence, we can rewrite the joint probability of the data, also referred to as the likelihood function, given the parameters $\boldsymbol{\pi}$, $\boldsymbol{\mu}, \boldsymbol{\Sigma}, \boldsymbol{\gamma}, \boldsymbol{\beta}, k$ as

$$
p\left(\boldsymbol{x}, t;\ \boldsymbol{\pi}, \boldsymbol{\mu}, \boldsymbol{\Sigma}, \boldsymbol{\gamma}, \boldsymbol{\beta}, k\right) = \int_{\boldsymbol{z}} \sum_{c=1}^K p(\boldsymbol{x}|\boldsymbol{z};\ \boldsymbol{\gamma})p(t|\boldsymbol{z}, c;\ \boldsymbol{\beta}, k)p(\boldsymbol{z}|c;\ \boldsymbol{\mu}, \boldsymbol{\Sigma})p(c;\ \boldsymbol{\pi}),
\tag{2}
$$

where $\boldsymbol{\mu} = \{\boldsymbol{\mu}_1, ..., \boldsymbol{\mu}_K\}$, $\boldsymbol{\Sigma} = \{\boldsymbol{\Sigma}_1, ..., \boldsymbol{\Sigma}_K\}$, and $\boldsymbol{\beta} = \{\boldsymbol{\beta}_1, ..., \boldsymbol{\beta}_K\}$.

**Evidence Lower Bound**  Given the data generating assumptions stated before, the objective is to infer the parameters $\boldsymbol{\pi}$, $\boldsymbol{\mu}$, $\boldsymbol{\Sigma}$, $\boldsymbol{\gamma}$, and $\boldsymbol{\beta}$ which better explain the covariates and survival outcomes $\{\boldsymbol{x}_i, t_i\}_{i=1}^N$. Since the likelihood function in Equation 2 is intractable, we maximise a lower bound of the log marginal probability of the data:

$$
\log p(\boldsymbol{x}, t; \boldsymbol{\pi}, \boldsymbol{\mu}, \boldsymbol{\Sigma}, \boldsymbol{\gamma}, \boldsymbol{\beta}, k) \geq \mathbb{E}_{q(\boldsymbol{z}, c|\boldsymbol{x}, t)} \log\left[\frac{p(\boldsymbol{x}|\boldsymbol{z}; \boldsymbol{\gamma})p(t|\boldsymbol{z}, c; \boldsymbol{\beta}, k)p(\boldsymbol{z}|c; \boldsymbol{\mu}, \boldsymbol{\Sigma})p(c; \boldsymbol{\pi})}{q(\boldsymbol{z}, c|\boldsymbol{x}, t)}\right].
\tag{3}
$$

We approximate the probability of the latent variables $\boldsymbol{z}$ and $c$ given the observations with a variational distribution $q(\boldsymbol{z}, c|\boldsymbol{x}, t) = q(\boldsymbol{z}|\boldsymbol{x})q(c|\boldsymbol{z}, t)$, where the first term is the encoder parameterised by a neural network. The second term is equal to the true probability $p(c|\boldsymbol{z}, t)$:

$$
q(c|\boldsymbol{z}, t) = p(c|\boldsymbol{z}, t) = \frac{p(\boldsymbol{z}, t|c)p(c)}{\sum_{c=1}^K p(\boldsymbol{z}, t|c)p(c)} = \frac{p(t|\boldsymbol{z}, c)p(\boldsymbol{z}|c)p(c)}{\sum_{c=1}^K p(t|\boldsymbol{z}, c)p(\boldsymbol{z}|c)p(c)}.
\tag{4}
$$

Thus, the evidence lower bound (ELBO) can be written as

$$
\begin{aligned}
\mathcal{L}(\boldsymbol{x}, t) = &\mathbb{E}_{q(\boldsymbol{z}|\boldsymbol{x})p(c|\boldsymbol{z}, t)} \log p(\boldsymbol{x}|\boldsymbol{z};\ \boldsymbol{\gamma}) + \mathbb{E}_{q(\boldsymbol{z}|\boldsymbol{x})p(c|\boldsymbol{z}, t)} \log p(t|\boldsymbol{z}, c;\ \boldsymbol{\beta}, k) \\
&- D_{\text{KL}}\left(q\left(\boldsymbol{z}, c|\boldsymbol{x}, t\right) \| p\left(\boldsymbol{z}, c;\ \boldsymbol{\mu}, \boldsymbol{\Sigma}, \boldsymbol{\pi}\right)\right).
\end{aligned}
\tag{5}
$$

Of particular interest is the second term which encourages the model to maximise the probability of observing the given survival outcome $t$ under the variational distribution of the latent embeddings and cluster assignments $q(\boldsymbol{z}, c|\boldsymbol{x}, t)$. It can be then seen as a mixture of survival distributions, each one assigned to one cluster. The ELBO can be approximated using the stochastic gradient variational Bayes (SGVB) estimator (Kingma & Welling, 2014) to be maximised efficiently using stochastic gradient descent. For the complete derivation, we refer to Appendix D.

**Missing Survival Time**  The hard cluster assignments can be computed from the distribution $p(c|\boldsymbol{z}, t)$ of Equation 4. However, the survival times may not be observable at test-time; whereas our derivation of the distribution $p(c|\boldsymbol{z}, t)$ depends on $p(t|\boldsymbol{z}, c)$. Therefore, when the survival time of an individual is unknown, using the Bayes' rule we instead compute $p(c|\boldsymbol{z}) = \frac{p(\boldsymbol{z}|c)p(c)}{\sum_{c=1}^K p(\boldsymbol{z}|c)p(c)}$.

## 4 EXPERIMENTAL SETUP

**Datasets**   We evaluate VaDeSC on a range of synthetic, semi-synthetic (survMNIST; Pölsterl (2019)), and real-world survival datasets with varying numbers of data points, explanatory variables, and fractions of censored observations (see Table 2). In particular, real-world clinical datasets include two benchmarks common in the survival analysis literature, namely SUPPORT (Knaus et al., 1995) and FLChain (Kyle et al., 2006; Dispenzieri et al., 2012); an observational cohort of pediatric patients undergoing chronic hemodialysis (Hemodialysis; Gotta et al. (2021)); an observational cohort of high-grade glioma patients (HGG); and an aggregation of several computed tomography (CT) image datasets acquired from patients diagnosed with non-small cell lung cancer (NSCLC; Aerts et al. (2019); Bakr et al. (2017); Clark et al. (2013); Weikert et al. (2019)). Detailed description of the datasets and preprocessing can be found in Appendices E and G. For (semi-)synthetic data, we focus on the clustering performance of the considered methods; whereas for real-world data, where the true cluster structure is unknown, we compare time-to-event predictions. In addition, we provide an in-depth cluster analysis for the NSCLC (see Section 5.3) and Hemodialysis (see Appendix H.8) datasets.

Table 2: Summary of the datasets. Here, $N$ is the total number of data points, $D$ is the number of explanatory variables, $K$ is the number of clusters if known. We report the percentage of censored observations and whether the cluster sizes are balanced if known.

| Dataset | $N$ | $D$ | % censored | Data type | $K$ | Balanced? | Section |
|---|---|---|---|---|---|---|---|
| Synthetic | 60,000 | 1,000 | 30 | Tabular | 3 | Y | 5.1, H.2 |
| survMNIST | 70,000 | 28×28 | 52 | Image | 5 | N | 5.1, H.6 |
| SUPPORT | 9,105 | 59 | 32 | Tabular | — | — | 5.2 |
| FLChain | 6,524 | 7 | 70 | Tabular | — | — | H.7 |
| HGG | 453 | 147 | 25 | Tabular | — | — | 5.2 |
| Hemodialysis | 1,493 | 57 | 91 | Tabular | — | — | 5.2, H.8 |
| NSCLC | 961 | 64×64 | 33 | Image | — | — | 5.3, H.9 |

**Baselines & Ablations**   We compare our method to several well-established baselines: the semi-supervised clustering (SSC; Bair & Tibshirani (2004)), survival cluster analysis (Chapfuwa et al., 2020), and deep survival machines (Nagpal et al., 2021a). For the sake of fair comparison, in SCA we truncate the Dirichlet process at the true number of clusters if known. For all neural network techniques, we use the same encoder architectures and numbers of latent dimensions. Although the profile regression approach of Liverani et al. (2020) is closely related to ours, it is not scalable to large unstructured datasets, such as survMNIST and NSCLC, since it relies on MCMC methods for Bayesian inference and is not parameterised by neural networks. Therefore, a full-scale comparison is impossible due to computational limitations. Appendix H.3 contains a 'down-scaled' experiment with the profile regression on synthetic data. Additionally, we consider $k$-means and regularised Cox PH and Weibull AFT models (Simon et al., 2011) as naïve baselines. For the VaDeSC, we perform several ablations: (*i*) removing the Gaussian mixture prior and performing *post hoc* $k$-means clustering on latent representations learnt by a VAE with an auxiliary Weibull survival loss term, which is similar to the deep survival analysis (DSA; Ranganath et al. (2016)) combined with $k$-means; (*ii*) training a completely unsupervised version without modelling the survival, which is similar to VaDE (Jiang et al., 2017); and (*iii*) predicting cluster assignments when the survival time is unobserved. Appendix G contains further implementation details.

**Evaluation**   We evaluate the clustering performance of models, when possible, in terms of accuracy (ACC), normalised mutual information (NMI), and adjusted Rand index (ARI). For the time-to-event predictions, we evaluate the ability of methods to rank individuals by their risk using the concordance index (CI; Raykar et al. (2007)). Predicted median survival times are evaluated using the relative absolute error (RAE; Yu et al. (2011)) and calibration slope (CAL), as implemented by Chapfuwa et al. (2020). We report RAE on both non-censored ($\text{RAE}_{nc}$) and censored ($\text{RAE}_c$) data points (see Equations 14 and 15 in Appendix F). The relative absolute error quantifies the relative deviation of median predictions from the observed survival times; while the calibration slope indicates whether a model tends to under- or overestimate risk on average. For the (semi-)synthetic datasets we average all results across independent *simulations*, *i.e.* dataset replicates; while for the real-world data we use the Monte Carlo cross-validation procedure.

## 5 RESULTS

### 5.1 CLUSTERING

We first compare clustering performance on the nonlinear (semi-)synthetic data. Table 3 shows the results averaged across several simulations. In addition to the clustering, we evaluate the concordance index to verify that the methods can adequately model time-to-event in these datasets. Training set results are reported in Table 12 in Appendix H.

As can be seen, in both problems, VaDeSC outperforms other models in terms of clustering and achieves performance comparable to SCA and DSM w.r.t. the CI. Including survival times appears to help identify clusters since the completely unsupervised VaDE achieves significantly worse results. It is also assuring that the model is able to predict clusters fairly well even when the survival time is not given at test time ("w/o $t$"). Furthermore, performing $k$-means clustering in the latent space of a VAE with the Weibull survival loss ("VAE + Weibull") clearly does not lead to the identification of the correct clusters. In both datasets, $k$-means on VAE representations yields almost no improvement over $k$-means on raw features. This suggests that the Gaussian mixture structure incorporated in the generative process of VaDeSC plays an essential role in inferring clusters.

Interestingly, while SCA and DSM achieve good results on survMNIST, both completely fail to identify clusters correctly on synthetic data, for which the generative process is very similar to the one assumed by VaDeSC. In the synthetic data, the clusters do not have prominently different survival distributions (see Appendix E.1); they are rather characterised by different associations between the covariates and survival times — whereas the two baseline methods tend to discover clusters with disparate survival distributions. The SSC offers little to no gain over the conventional $k$-means performed on the complete feature set. Last, we note that in both datasets VaDeSC has a significantly better CI than the Cox PH model likely due to its ability to capture nonlinear relationships between the covariates and outcome.

Figure 4 provides a closer inspection of the clustering and latent representations on survMNIST data. It appears that SCA and DSM, as expected, fail to discover clusters with similar Kaplan–Meier (KM) curves and have a latent space that is driven purely by survival time. While the VAE + Weibull model learns representations driven by both the explanatory variables (digits in the images) and survival time, the *post hoc* $k$-means clustering fails at identifying the true clusters. By contrast, VaDeSC is capable of discovering clusters with even minor differences in KM curves and learns

Table 3: Test set clustering performance on synthetic and survMNIST data. "VAE + Weibull" corresponds to an ablation of VaDeSC w/o the Gaussian mixture prior. "w/o $t$" corresponds to the cluster assignments made by VaDeSC when the survival time is not given. Averages and standard deviations are reported across 5 and 10 independent simulations, respectively. Best results are shown in **bold**, second best – in *italic*.

| Dataset | Method | ACC | NMI | ARI | CI |
|---|---|---|---|---|---|
| Synthetic | $k$-means | 0.44±0.04 | 0.06±0.04 | 0.05±0.03 | — |
| | Cox PH | — | — | — | 0.77±0.02 |
| | SSC | 0.45±0.03 | 0.08±0.04 | 0.06±0.02 | — |
| | SCA | 0.45±0.09 | 0.05±0.05 | 0.04±0.05 | *0.82±0.02* |
| | DSM | 0.37±0.02 | 0.01±0.00 | 0.01±0.00 | 0.76±0.02 |
| | VAE + Weibull | 0.46±0.06 | 0.09±0.04 | 0.09±0.04 | 0.71±0.02 |
| | VaDE | 0.74±0.21 | 0.53±0.12 | 0.55±0.20 | — |
| | VaDeSC (w/o $t$) | *0.88±0.03* | *0.60±0.07* | *0.67±0.07* | **0.84±0.02** |
| | VaDeSC (ours) | **0.90±0.02** | **0.66±0.05** | **0.73±0.05** | |
| survMNIST | $k$-means | 0.49±0.06 | 0.31±0.04 | 0.22±0.04 | — |
| | Cox PH | — | — | — | 0.74±0.04 |
| | SSC | 0.49±0.06 | 0.31±0.04 | 0.22±0.04 | — |
| | SCA | 0.56±0.09 | 0.46±0.06 | 0.33±0.10 | *0.79±0.06* |
| | DSM | 0.54±0.11 | 0.40±0.16 | 0.31±0.14 | *0.79±0.05* |
| | VAE + Weibull | 0.49±0.05 | 0.32±0.05 | 0.24±0.05 | 0.76±0.07 |
| | VaDE | 0.47±0.07 | 0.38±0.08 | 0.24±0.08 | — |
| | VaDeSC (w/o $t$) | *0.57±0.09* | *0.51±0.09* | *0.37±0.10* | **0.80±0.05** |
| | VaDeSC (ours) | **0.58±0.10** | **0.55±0.11** | **0.39±0.11** | |

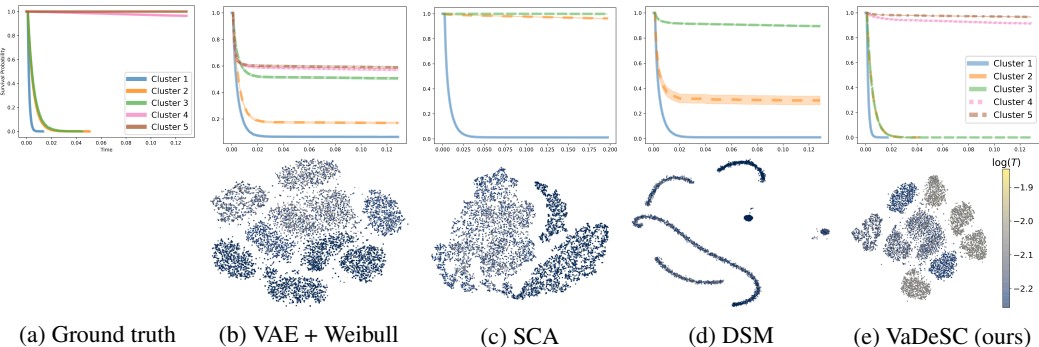

(a) Ground truth     (b) VAE + Weibull     (c) SCA     (d) DSM     (e) VaDeSC (ours)

Figure 4: Cluster-specific Kaplan–Meier (KM) curves (*top*) and $t$-SNE visualisation of latent representations (*bottom*), coloured according to survival times (yellow and blue correspond to higher and lower survival times, respectively), learnt by different models (b-e) from one replicate of the survMNIST dataset. Panel (a) shows KM curves of the ground truth clusters. Plots were generated using 10,000 data points randomly sampled from the training set; similar results were observed on the test set.

Table 4: Test set time-to-event performance on SUPPORT, HGG, and Hemodialysis datasets. Averages and standard deviations are reported across 5 independent train-test splits.

| Dataset | Method | CI | $RAE_{nc}$ | $RAE_c$ | CAL |
|---|---|---|---|---|---|
| SUPPORT | Cox PH | 0.84±0.01 | — | — | — |
| | Weibull AFT | 0.84±0.01 | 0.62±0.01 | *0.13±0.01* | *1.27±0.02* |
| | SCA | 0.83±0.02 | 0.78±0.13 | **0.06±0.04** | 1.74±0.52 |
| | DSM | **0.87±0.01** | *0.56±0.02* | *0.13±0.04* | 1.43±0.07 |
| | VAE + Weibull | 0.84±0.01 | *0.56±0.02* | 0.20±0.02 | 1.28±0.04 |
| | VaDeSC (ours) | *0.85±0.01* | **0.53±0.02** | 0.23±0.05 | **1.24±0.05** |
| HGG | Cox PH | *0.74±0.05* | — | — | — |
| | Weibull AFT | *0.74±0.05* | 0.56±0.04 | 0.14±0.09 | 1.16±0.10 |
| | SCA | 0.63±0.08 | 0.97±0.05 | **0.00±0.00** | 2.59±1.70 |
| | DSM | **0.75±0.04** | 0.57±0.05 | 0.18±0.07 | **1.09±0.08** |
| | VAE + Weibull | **0.75±0.05** | **0.52±0.06** | *0.12±0.07* | 1.14±0.11 |
| | VaDeSC (ours) | *0.74±0.05* | *0.53±0.06* | 0.13±0.07 | *1.12±0.09* |
| Hemodialysis | Cox PH | **0.83±0.04** | — | — | — |
| | Weibull AFT | **0.83±0.05** | 0.81±0.03 | **0.01±0.00** | *4.46±0.59* |
| | SCA | 0.75±0.05 | 0.86±0.07 | *0.02±0.02* | 7.93±3.22 |
| | DSM | *0.80±0.06* | 0.85±0.08 | *0.02±0.04* | 8.23±4.28 |
| | VAE + Weibull | 0.77±0.06 | *0.80±0.06* | *0.02±0.01* | 4.49±0.75 |
| | VaDeSC (ours) | *0.80±0.05* | **0.78±0.05** | **0.01±0.00** | **3.74±0.58** |

representations that clearly reflect the covariate and survival variability. Similar differences can be observed for the synthetic data (see Appendix H.2).

## 5.2   TIME-TO-EVENT PREDICTION

We now assess time-to-event prediction on clinical data. As these datasets do not have ground truth clustering labels, we do not evaluate the clustering performance. However, for Hemodialysis, we provide an in-depth qualitative assessment of the learnt clusters in Appendix H.8.

Table 4 shows the time-to-event prediction performance on SUPPORT, HGG, and Hemodialysis. Results for FLChain are reported in Appendix H.7 and yield similar conclusions. Surprisingly, SCA often has a considerable variance w.r.t. the calibration slope, sometimes yielding badly calibrated predictions. Note, that for SCA and DSM, the results differ from those reported in the original papers likely due to a different choice of encoder architectures and numbers of latent dimensions. In general, these results are promising and suggest that VaDeSC remains competitive at time-to-event modelling, offers overall balanced predictions, and is not prone to extreme overfitting even when applied to simple clinical datasets that are low-dimensional or contain few non-censored patients.

## 5.3 APPLICATION TO COMPUTED TOMOGRAPHY DATA

We further demonstrate the viability of our model in a real-world application using a collection of several CT image datasets acquired from NSCLC patients (see Appendix E). The resulting dataset poses two major challenges. The first one is the high variability among samples, mainly due to disparate lung tumour locations. In addition, several demographic characteristics, such as patient's age, sex, and weight, might affect both the CT scans and survival outcome. Thus, a representation learning model that captures cluster-specific associations between medical images and survival times could be highly beneficial to both explore the sources of variability in this dataset and to understand the different generative mechanisms of the survival outcome. The second challenge is the modest dataset size ($N = 961$). To this end, we leverage image augmentation during neural network training (Perez & Wang, 2017) to mitigate spurious associations between survival outcomes and clinically irrelevant CT scan characteristics (see Appendix G).

We compare our method to the DSM, omitting the SCA, which seems to yield results similar to the latter. As well-established time-to-event prediction baselines, we fit Cox PH and Weibull AFT models on *radiomics features*. The latter are extracted using manual pixel-wise tumour segmentation maps, which are time-consuming and expensive to obtain. Note that neither DSM nor VaDeSC requires segmentation as an input. Table 5 shows the time-to-event prediction results. Interestingly, DSM and VaDeSC achieve performance comparable to the radiomics-based models, even yielding a slightly better calibration on average. This suggests that laborious manual tumour segmentation is not necessary for survival analysis on CT scans. Similarly to tabular clinical datasets (see Table 4), DSM and VaDeSC have comparable performance. Therefore, we investigate cluster assignments qualitatively to highlight the differences between the two methods.

Table 5: Test set time-to-event prediction performance on NSCLC data. For Cox PH and Weibull AFT, radiomics features were extracted using tumour segmentation maps. Averages and standard deviations are reported across 100 independent train-test splits, stratified by survival time.

| Method | CI | $RAE_{nc}$ | $RAE_c$ | CAL |
|---|---|---|---|---|
| Radiomics + Cox PH | **0.60±0.02** | — | — | — |
| Radiomics + Weibull AFT | **0.60±0.02** | **0.70±0.02** | 0.45±0.03 | 1.26±0.04 |
| *DSM* | *0.59±0.04* | *0.72±0.03* | ***0.34±0.06*** | *1.24±0.07* |
| *VaDeSC (ours)* | ***0.60±0.02*** | *0.71±0.03* | *0.35±0.05* | ***1.21±0.05*** |

In Figure 5, we plot the KM curves and corresponding centroid CT images for the clusters discovered by DSM and VaDeSC on one train-test split. By performing several independent experiments (see Appendix H.10), we observe that both methods discover patient groups with different empirical survival time distributions. However, in contrast to DSM, VaDeSC clusters are consistently associated with the tumour location. On the contrary, the centroid CT images of the DSM clusters show

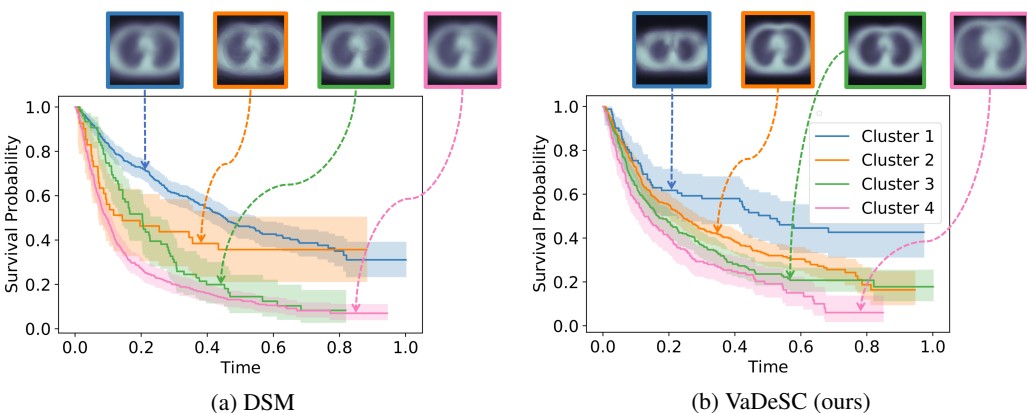

(a) DSM          (b) VaDeSC (ours)

Figure 5: Cluster-specific Kaplan–Meier curves and corresponding centroid CT images, computed by averaging all samples assigned to each cluster by (a) DSM and (b) VaDeSC on the NSCLC data.

Table 6: Cluster-specific statistics for a few demographic and clinical variables (not used during training) from the NSCLC dataset for DSM and VaDeSC. T. Vol. stands for the tumour volume; M1 denotes spread of cancer to distant organs and tissues; and $\geq$ T3 denotes a tumour stage of at least 3. Kruskal-Wallis $H$-test $p$-values are reported at the significance levels of 0.001, 0.01, and 0.05.

| Variable | DSM | | | | | VaDeSC (ours) | | | | |
|---|---|---|---|---|---|---|---|---|---|---|
| | 1 | 2 | 3 | 4 | $p$-val. | 1 | 2 | 3 | 4 | $p$-val. |
| **T. Vol.**, $cm^3$ | 23 | 39 | 38 | 51 | $\leq$ **1e-3** | 43 | 36 | 40 | 63 | $\leq$ **5e-2** |
| **Age**, yrs | 67 | 68 | 68 | 69 | 0.11 | 62 | 69 | 67 | 70 | $\leq$ **1e-3** |
| **Female**, % | 29 | 30 | 26 | 21 | 0.3 | 36 | 19 | 38 | 23 | $\leq$ **1e-3** |
| **Smoker**, % | 84 | 100 | 80 | 89 | 0.9 | 67 | 94 | 87 | 100 | 0.12 |
| **M1**, % | 40 | 55 | 16 | 42 | 0.4 | 20 | 45 | 44 | 45 | 0.2 |
| $\geq$ **T3**, % | 27 | 12 | 23 | 32 | 0.2 | 10 | 29 | 35 | 31 | 0.7 |

no visible difference. This is further demonstrated in Figure 6, where we generated several CT scans for each cluster by sampling from the multivariate Gaussian distribution in the latent space of the VaDeSC. We observe a clear association with tumour location, as clusters 1 (■) and 3 (■) correspond to the upper section of the lungs. Indeed, multiple studies have suggested a higher five-year survival rate in patients with the tumour in the upper lobes (Lee et al., 2018). It is also interesting to observe the amount of variability within each cluster: every generated scan is characterised by a unique lung shape and tumour size. Since DSM is not a generative model, we instead plot the original samples assigned to each cluster by DSM in Appendix H.9. Finally, in Table 6 we compute cluster-specific statistics for a few important demographic and clinical variables (Etiz et al., 2002; Agarwal et al., 2010; Bhatt et al., 2014). We observe that VaDeSC tends to discover clusters with more disparate characteristics and is thus able to stratify patients not only by risk but also by clinical conditions. Overall, the results above agree with our findings on (semi-)synthetic data: VaDeSC identifies clusters informed by both the covariates *and* survival outcomes, leading to a stratification very different from previous approaches.

**Mean**      **Generated Samples**

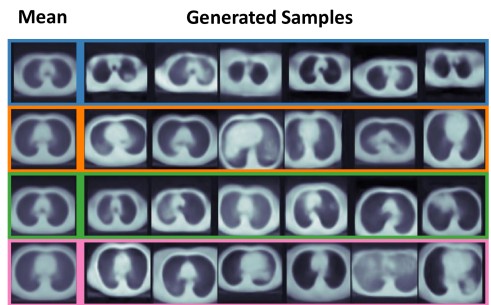

Figure 6: CT images generated by (*i*) sampling latent representations from the Gaussian mixture learnt by VaDeSC and (*ii*) decoding representations using the decoder network.

## 6 CONCLUSION

In this paper, we introduced a novel deep probabilistic model for clustering survival data. In contrast to existing approaches, our method can retrieve clusters driven by both the explanatory variables and survival information and it can be trained efficiently in the framework of stochastic gradient variational inference. Empirically, we showed that our model offers an improvement in clustering performance compared to the related work while staying competitive at time-to-event modelling. We also demonstrated that our method identifies meaningful clusters from a challenging medical imaging dataset. The analysis of these clusters provides interesting insights that could be useful for clinical decision-making. To conclude, the proposed VaDeSC model offers a holistic perspective on clustering survival data by learning structured representations which reflect the covariates, outcomes, and varying relationships between them.

**Limitations & Future Work** The proposed model has a few limitations. It requires fixing a number of mixture components *a priori* since global parameters are not treated in a fully Bayesian manner, as opposed to the profile regression. Although the obtained cluster assignments can be explained *post hoc*, the relationship between the raw features and survival outcomes remains unclear. Thus, further directions of work include (*i*) improving the interpretability of our model to facilitate its application in the medical domain and (*ii*) a fully Bayesian treatment of global parameters. In-depth interpretation of the clusters we found in clinical datasets is beyond the scope of this paper, but we plan to investigate these clusters together with our clinical collaborators in follow-up work.

## ACKNOWLEDGEMENTS

We thank Alexandru Țifrea, Nicolò Ruggeri, and Kieran Chin-Cheong for valuable discussion and comments and Dr. Silvia Liverani, Paidamoyo Chapfuwa, and Chirag Nagpal for sharing the code. Ričards Marcinkevičs is supported by the SNSF grant #320038189096. Laura Manduchi is supported by the PHRT SHFN grant #1-000018-057: SWISSHEART.

## ETHICS STATEMENT

The in-house PET/CT and HGG data were acquired at the University Hospital Basel and University Hospital Zürich, respectively, and retrospective, observational studies were approved by local ethics committees (Ethikkommission Nordwest- und Zentralschweiz, no. 2016-01649; Kantonale Ethikkommission Zürich, no. PB-2017-00093).

## REPRODUCIBILITY STATEMENT

To ensure the reproducibility of this work several measures were taken. The experimental setup is outlined in Section 4. Appendix E details benchmarking datasets and simulation procedures. Appendix F defines metrics used for model comparison. Appendix G contains data preprocessing, architecture, and hyperparameter tuning details. For the computed tomography data we assess the stability of the obtained results in Appendix H.10. Most datasets used for comparison are either synthetic or publicly available. HGG, Hemodialysis, and in-house PET/CT data could not be published due to medical confidentiality. The code is publicly available at https://github.com/i6092467/vadesc.

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

## A    CENSORING IN SURVIVAL ANALYSIS

*Censoring* is one of the key features of survival analysis that distinguish it from the classical regression modelling (Rodríguez, 2007). For some units in the study the event of interest, *e.g.* death, is observed, whereas for others it does not occur during the observation period. The latter units are referred to as *censored*, and the observation time for these units is a lower bound on the time-to-event. The censoring event described above is commonly known as *right censoring* (Winkel, 2007) and is the only form of censoring considered throughout the current paper. Figure 7 depicts an example of the survival data: 30 patients from an observational cohort, 13 of whom were censored. Although there exist several types of censoring, most of the schemes assume that it is *non-informative*, *i.e.* that the censoring time is independent of unit's survival beyond it (Rodríguez, 2007). This assumption is made by the VaDeSC model as well.

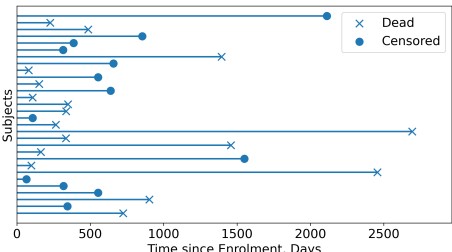

Figure 7: Survival time, in days, measured from the enrolment in the study for a few subjects from an observational cohort of high-grade glioma patients. Here, × corresponds to death (the event of interest), and ● denotes censoring.

## B    RELATED WORK

Herein, we provide a detailed overview of the related work. The proposed VaDeSC model (see Section 3) builds on three major topics: (*i*) probabilistic deep unsupervised clustering, (*ii*) mixtures of regression models, and (*iii*) survival analysis.

**Deep Unsupervised Clustering**    Long-established clustering algorithms, such as $k$-means and Gaussian Mixture Models, have been recently combined with deep neural networks to learn better representations of high-dimensional data (Min et al., 2018). Several techniques have been proposed in the literature, featuring a wide range of neural network architectures. Among them, the VAE (Kingma & Welling, 2014; Rezende et al., 2014) combines variational Bayesian methods with the flexibility and scalability of neural networks. Gaussian mixture variational autoencoders (GMM-VAE; Dilokthanakul et al. (2016)) and variational deep embedding (VaDE; Jiang et al. (2017)) are variants of the VAE in which the prior is a Gaussian mixture distribution. With this assumption, the data is clustered in the latent space of the VAE and the resulting inference model can be directly optimised within the framework of stochastic gradient variational Bayes. Further extensions focus on the inclusion of side information in the form of *pairwise constraints* (Manduchi et al., 2021), overcoming the restrictive i.i.d. assumption of the samples, or on a fully Bayesian treatment of global parameters (Luo et al., 2018; Johnson et al., 2016), alleviating the need to fix the number of clusters *a priori*.

**Mixtures of Regression Models**    Mixtures of regression (McLachlan & Peel, 2004) model a response variable by a mixture of individual regressions with the help of latent cluster assignments. Such mixtures do not have to be limited to a finite number of components; for example, profile regression (Molitor et al., 2010) leverages the Dirichlet mixture model for cluster assignment. Along similar lines, within the machine learning community, mixture of experts models (MoE) were introduced by Jacobs et al. (1991).

Mixture models have been successfully leveraged for survival analysis as well (Farewell, 1982; Rosen & Tanner, 1999; Liverani et al., 2020; Chapfuwa et al., 2020; Nagpal et al., 2021a;b). For instance, Farewell (1982) considered fitting separate models for two subpopulations: short-term and long-term survivors. Rosen & Tanner (1999) extended the classical Cox PH model with the MoE. Recent neural network approaches typically fit mixtures of survival models on rich representations produced by an encoding neural network (Chapfuwa et al., 2020; Nagpal et al., 2021a;b).

**Nonlinear Survival Analysis**    One of the first machine learning approaches for survival analysis was Faraggi–Simon's network (Faraggi & Simon, 1995), which was an extension of the classical Cox PH model (Cox, 1972). Since then, an abundance of machine learning methods have been

developed: random survival forests (Ishwaran et al., 2008), deep survival analysis (Ranganath et al., 2016), neural networks based on the Cox PH model (Katzman et al., 2018; Kvamme et al., 2019), adversarial time-to-event modelling (Chapfuwa et al., 2018), and deep survival machines (Nagpal et al., 2021a) etc. An exhaustive overview of these and other techniques is beyond the scope of the current paper.

## C  COMPARISON WITH RELATED WORK

**Generative Assumptions**    In addition to the differences summarised in Table 1, VaDeSC and the related approaches (Chapfuwa et al., 2020; Nagpal et al., 2021a; Liverani et al., 2020) assume different data generating processes. Simplified graphical models assumed by these techniques are shown in Figure 8. Both SCA and DSM are not generative w.r.t. explanatory variables $x$, and thus, do not maximise the joint likelihood $\mathcal{L}(x, t)$ and are outcome-driven. On the other hand, both profile regression and VaDeSC specify a model for covariates $x$, as well as for survival time $t$. However, PR does not infer latent representations $z$ and importantly, in its simplest form assumes that $x \perp\!\!\!\perp t | c$. The latter assumption can limit the discriminative power of a mixture with few components. By default, VaDeSC does not make such restrictive assumptions.

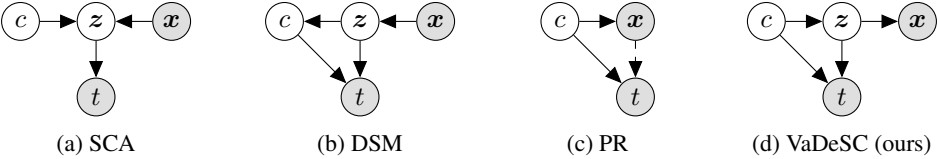

|  |  |  |  |
|---|---|---|---|
| (a) SCA | (b) DSM | (c) PR | (d) VaDeSC (ours) |

Figure 8: Comparison of the generative assumptions made by the models related to the VaDeSC: (a) survival cluster analysis (SCA; Chapfuwa et al. (2020)), (b) deep survival machines (DSM; Nagpal et al. (2021a)), (c) and profile regression for survival data (PR; Liverani et al. (2020)). For the sake of brevity, the censoring indicator $\delta$, model parameters and hyperparameters were omitted. Here, $t$ denotes survival time, $x$ denotes explanatory variables, $z$ corresponds to the latent representation, and $c$ stands for the cluster assignment. Shaded nodes correspond to observed variables. The presence of dashed edges depends on the modelling choice.

**Deep Cox Mixtures**    The concurrent work by Nagpal et al. (2021b) on deep Cox mixtures allows leveraging latent representations learnt by a VAE to capture nonlinear relationships in the data. The hidden representation is simultaneously used to fit a mixture of $K$ Cox models. In contrast to our method, DCM does not specify a generative model explicitly. The VAE loss is added as a weighted term to the main loss derived from the Cox models and is optimised jointly using a Monte Carlo EM algorithm. The VAE is mainly used to enforce a Gaussian prior in the latent space, resulting in more compact representations. It can be replaced with a regularised AE, or even an MLP encoder, as mentioned by the authors (Nagpal et al., 2021b). Thus, the maximisation of the joint likelihood $\mathcal{L}(x, t)$, rather than of $\mathcal{L}(t)$, is not the main emphasis of the DCMs. On the contrary, our approach is probabilistic and specifies a clear and interpretable generative process from which an ELBO of the joint likelihood can be derived formally. Additionally, VaDeSC enforces a more structured representation in the latent space, since it uses a Gaussian mixture prior.

**Empirical Contribution**    Last but not least, the experimental setup of the current paper is different from those of Bair & Tibshirani (2004); Chapfuwa et al. (2020); Liverani et al. (2020); Nagpal et al. (2021a;b). Previous work on mixture modelling and clustering for survival data has focused on applications to tabular datasets. Survival analysis on image datasets (Haarburger et al., 2019; Bello et al., 2019) poses a challenge pertinent to many application areas, *e.g.* medical imaging. An important empirical contribution of this work is to apply the proposed technique and demonstrate its utility on computed tomography data (see Section 5.3).

## D    ELBO TERMS

In this appendix, we provide the complete derivation of the ELBO of VaDeSC using the SGVB estimator. From Equation 5 we can rewrite the ELBO as

$$
\mathcal{L}(\boldsymbol{x}, t) = \mathbb{E}_{q(\boldsymbol{z}|\boldsymbol{x})p(c|\boldsymbol{z},t)} \big[ \log p(\boldsymbol{x}|\boldsymbol{z};\ \boldsymbol{\gamma}) + \log p(t|\boldsymbol{z}, c;\ \boldsymbol{\beta}, k) + \log p(\boldsymbol{z}|c;\ \boldsymbol{\mu}, \boldsymbol{\Sigma})
$$
$$
+ \log p(c;\ \boldsymbol{\pi}) - \log q(\boldsymbol{z}, c|\boldsymbol{x}, t) \big].
\tag{6}
$$

In the following we investigate each term of the equation above.

**Reconstruction**    The first term is also known as the reconstruction term in a classical VAE setting. Herein, we provide the derivation for $p(\boldsymbol{x}|\boldsymbol{z};\ \boldsymbol{\gamma}) = \text{Bernoulli}(\boldsymbol{x};\ \boldsymbol{\mu_\gamma})$, *i.e.* for binary-valued explanatory variables:

$$
\mathbb{E}_{q(\boldsymbol{z}|\boldsymbol{x})p(c|\boldsymbol{z},t)} \log p(\boldsymbol{x}|\boldsymbol{z};\ \boldsymbol{\gamma}) = \mathbb{E}_{q(\boldsymbol{z}|\boldsymbol{x})} \log p(\boldsymbol{x}|\boldsymbol{z};\ \boldsymbol{\gamma}) \approx \frac{1}{L} \sum_{l=1}^{L} \log p(\boldsymbol{x}|\boldsymbol{z}^{(l)};\ \boldsymbol{\gamma})
$$
$$
= \frac{1}{L} \sum_{l=1}^{L} \sum_{d=1}^{D} x_d \log \mu_{\gamma,d}^{(l)} + (1 - x_d) \log(1 - \mu_{\gamma,d}^{(l)}),
\tag{7}
$$

where $L$ is the number of Monte Carlo samples in the SGVB estimator; $\boldsymbol{\mu}_{\gamma}^{(l)} = f(\boldsymbol{z}^{(l)};\ \boldsymbol{\gamma})$; $\boldsymbol{z}^{(l)} \sim q(\boldsymbol{z}|\boldsymbol{x}) = \mathcal{N}(\boldsymbol{z};\ \boldsymbol{\mu_\theta}, \boldsymbol{\sigma_\theta^2})$; $[\boldsymbol{\mu_\theta}, \log \boldsymbol{\sigma_\theta^2}] = g(\boldsymbol{x};\ \boldsymbol{\theta})$ and $g(\cdot;\ \boldsymbol{\theta})$ is an encoder neural network parameterised by $\boldsymbol{\theta}$; $x_d$ refers to the $d$-th component of $\boldsymbol{x}$ and $D$ is the number of features. For other data types, the distribution $p(\boldsymbol{x}|\boldsymbol{z};\ \boldsymbol{\gamma})$ in Equation 7 needs to be chosen accordingly.

**Survival**    The second term of the ELBO includes the survival time. Following the survival model described by Equation 1, the SGVB estimator for the term is given by

$$
\mathbb{E}_{q(\boldsymbol{z}|\boldsymbol{x})p(c|\boldsymbol{z},t)} \log p(t|\boldsymbol{z}, c;\ \boldsymbol{\beta}, k) \approx \sum_{l=1}^{L} \sum_{\nu=1}^{K} \log p(t|\boldsymbol{z}^{(l)}, c = \nu;\ \boldsymbol{\beta}, k) p(c = \nu|\boldsymbol{z}^{(l)}, t)
$$
$$
= \frac{1}{L} \sum_{l=1}^{L} \sum_{\nu=1}^{K} \log \left[ \left\{ \frac{k}{\lambda_\nu^{\boldsymbol{z}^{(l)}}} \left( \frac{t}{\lambda_\nu^{\boldsymbol{z}^{(l)}}} \right)^{k-1} \right\}^{\delta} \exp\left( -\left( \frac{t}{\lambda_\nu^{\boldsymbol{z}^{(l)}}} \right)^k \right) \right] p(c = \nu|\boldsymbol{z}^{(l)}, t),
\tag{8}
$$

where $\lambda_\nu^{\boldsymbol{z}^{(l)}} = \text{softplus}\left( \boldsymbol{z}^{(l)\top} \boldsymbol{\beta}_\nu \right).$

**Clustering**    The third term in Equation 6 corresponds to the clustering loss and is defined as

$$
\mathbb{E}_{q(\boldsymbol{z}|\boldsymbol{x})p(c|\boldsymbol{z},t)} \log p(\boldsymbol{z}|c;\ \boldsymbol{\mu}, \boldsymbol{\Sigma}) = \mathbb{E}_{q(\boldsymbol{z}|\boldsymbol{x})} \sum_{\nu=1}^{K} p(c = \nu|\boldsymbol{z}, t) \log p(\boldsymbol{z}|c = \nu;\ \boldsymbol{\mu}, \boldsymbol{\Sigma})
$$
$$
\approx \frac{1}{L} \sum_{l=1}^{L} \sum_{\nu=1}^{K} p(c = \nu|\boldsymbol{z}^{(l)}, t) \log p(\boldsymbol{z}^{(l)}|c = \nu;\ \boldsymbol{\mu}, \boldsymbol{\Sigma}).
\tag{9}
$$

**Prior**    The fourth term can be approximated as

$$
\mathbb{E}_{q(\boldsymbol{z}|\boldsymbol{x})p(c|\boldsymbol{z},t)} \log p(c;\ \boldsymbol{\pi}) \approx \frac{1}{L} \sum_{l=1}^{L} \sum_{\nu=1}^{K} p(c = \nu|\boldsymbol{z}^{(l)}, t) \log p(c = \nu;\ \boldsymbol{\pi}).
\tag{10}
$$

**Variational**    Finally, the fifth term of the ELBO corresponds to the entropy of the variational distribution $q(\boldsymbol{z}, c|\boldsymbol{x}, t)$:

$$
-\mathbb{E}_{q(\boldsymbol{z}|\boldsymbol{x})p(c|\boldsymbol{z},t)} \log q(\boldsymbol{z}, c|\boldsymbol{x}, t) = \mathbb{E}_{q(\boldsymbol{z}|\boldsymbol{x})p(c|\boldsymbol{z},t)} [\log q(\boldsymbol{z}|\boldsymbol{x}) + \log p(c|\boldsymbol{z}, t)]
$$
$$
= -\mathbb{E}_{q(\boldsymbol{z}|\boldsymbol{x})} \log q(\boldsymbol{z}|\boldsymbol{x}) - \mathbb{E}_{q(\boldsymbol{z}|\boldsymbol{x})p(c|\boldsymbol{z},t)} \log p(c|\boldsymbol{z}, t),
\tag{11}
$$

where the first term is the entropy of a multivariate Gaussian distribution with a diagonal covariance matrix. We can then approximate the expression in Equation 11 by

$$\frac{J}{2}(\log(2\pi) + 1) + \sum_{j=1}^{J} \log \sigma_{\boldsymbol{\theta},j}^2 - \frac{1}{L}\sum_{l=1}^{L}\sum_{\nu=1}^{K} p(c = \nu | \boldsymbol{z}^{(l)}, t) \log p(c = \nu | \boldsymbol{z}^{(l)}, t), \qquad (12)$$

where $J$ is the number of latent space dimensions.

## E  DATASETS

Below we provide detailed summaries of the survival datasets used in the experiments (see Table 2).

**Synthetic**  As the simplest benchmark problem, we simulate synthetic survival data based on the generative process reminiscent of the VaDeSC generative model (see Figure 3). In this dataset, both covariates and survival times are generated based on latent variables which are distributed as a Gaussian mixture. The dependence between latent and explanatory variables is given by a multilayer perceptron and is, therefore, nonlinear. Appendix E.1 contains a detailed description of these simulations and a visualisation of one replicate (see Figure 9).

**survMNIST**  Another dataset we consider is the semi-synthetic survival MNIST (survMNIST) introduced by Pölsterl (2019), used for benchmarking nonlinear survival analysis models, *e.g.* by Goldstein et al. (2020). In survMNIST, explanatory variables are given by images of handwritten digits (LeCun et al., 2010). Digits are assigned to clusters and synthetic survival times are generated using the exponential distribution with a cluster-specific scale parameter. Appendix E.2 provides further details on the generation procedure.

**SUPPORT**  To compare models on real clinical data, we consider the Study to Understand Prognoses and Preferences for Outcomes and Risks of Treatments (SUPPORT; Knaus et al. (1995)) consisting of seriously ill hospitalised adults from five tertiary care academic centres in the United States. It includes demographic, laboratory, and scoring data acquired from patients diagnosed with cancer, chronic obstructive pulmonary disease, congestive heart failure, cirrhosis, acute renal failure, multiple organ system failure, and sepsis.

**FLChain**  A second clinical dataset was acquired in the course of the study of the relationship between serum free light chain (FLChain) and mortality (Kyle et al., 2006; Dispenzieri et al., 2012) conducted in Olmsted County, Minnesota in the United States. The dataset includes demographic and laboratory variables alongside with the recruitment year. FLChain is by far the most low-dimensional among the considered datasets (see Table 2). Both SUPPORT and FLChain data are publicly available.

**HGG**  Additionally, we perform experiments on an in-house dataset consisting of 453 patients diagnosed with high-grade glioma (HGG), a type of brain tumour (Buckner, 2003). The cohort includes patients with glioblastoma multiforme and anaplastic astrocytoma (Weller et al., 2020) treated surgically at the University Hospital Zürich between 03/2008 and 06/2017. The dataset consists of demographic variables, treatment assignments, pre- and post-operative volumetric variables, molecular markers, performance scores, histology findings, and information on tumour location. This dataset has by far the least amount of patients (see Table 2) and therefore, could be a good benchmark for the 'low data' regime.

**Hemodialysis**  Another clinical dataset we include is an observational cohort of patients from the DaVita Kidney Care (DaVita Inc., Denver, CO, USA) dialysis centres (Gotta et al., 2018; 2019a;b; 2020; 2021). The cohort is composed of patients who started chronic hemodialysis (HD) in childhood and have then received thrice-weekly HD between 05/2004 and 03/2016, with a maximum follow-up until $< 30$ years of age. The dataset consists of demographic factors, such as the age from the start of dialysis, gender, as well as etiology of kidney disease and comorbidities, dialysis dose in terms of spKt/V, eKt/V (Daugirdas, 1993; Daugirdas & Schneditz, 1995), fluid removal in terms of UFR (mL/kg/h) and total ultrafiltration (UF, % per kg dry weigth per session), and the interdialytic weight gain (IDWG) (Marsenic et al., 2016). It should be noted that this dataset is rather

challenging, given the high percentage of censored observations (see Table 2). The scientific use of the deidentified standardised electronic medical records was approved by DaVita (DaVita Clinical Research®, Minneapolis, MN); IRB approval was not required since retrospective analysis was performed on the deidentified dataset. A complete description of the variables and measurements used is provided by Gotta et al. (2021) in the *Methods* section.

**NSCLC** Finally, we perform experiments on a pooled dataset of five observational cohorts consisting of patients diagnosed with non-small cell lung cancer (NSCLC). We consider a large set of pretreatment CT scans and CT components of positron emission tomography (PET)/CT scans from the following sources:

- An in-house PET/CT dataset consisting of 392 patients treated at the University Hospital Basel (Weikert et al., 2019). It includes manual delineations, clinical and survival data. A retrospective, observational study was approved by the local ethics committee (Ethikkommission Nordwest- und Zentralschweiz, no. 2016-01649).

- Lung1 dataset consisting of 422 NSCLC patients treated at Maastro Clinic, the Netherlands (Aerts et al., 2014; 2019). For these patients, CT scans, manual delineations, clinical, and survival data are available. Lung1 is publicly available in The Cancer Imaging Archive (TCIA; Clark et al. (2013)).

- Lung3 dataset consisting of 89 NSCLC patients treated at Maastro Clinic, the Netherlands (Aerts et al., 2014; 2015). Lung3 includes pretreatment CT scans, tumour delineations, and gene expression profiles. It is is publicly available in TCIA.

- NSCLC Radiogenomics dataset acquired from 211 NSCLC patients from the Stanford University School of Medicine and Palo Alto Veterans Affairs Healthcare System (Bakr et al., 2017; 2018; Gevaert et al., 2012). The dataset comprises CT and PET/CT scans, segmentation maps, gene mutation and RNA sequencing data, clinical data, and survival outcomes. It is is publicly available in TCIA.

Only subjects with a tumour segmentation map and transversal CT or PET/CT scan available are retained. Patients who have the slice of (PET)/CT with the maximum tumour area outside the lungs, *e.g.* in the brain or abdomen, are excluded from the analysis. Thus, the final dataset includes 961 subjects.

## E.1 SYNTHETIC DATA GENERATION

Herein, we provide details on the generation of synthetic survival data used in our experiments (see Section 4). The procedure for simulating these data is very similar to the generative process assumed by the proposed VaDeSC model (*cf.* Figure 3) and can be summarised as follows:

1. Let $\pi_c = \frac{1}{K}$, for $1 \le c \le K$.

2. Sample $c_i \sim \text{Cat}(\boldsymbol{\pi})$, for $1 \le i \le N$.

3. Sample $\mu_{c,j} \sim \text{unif}\left(-\frac{1}{2}, \frac{1}{2}\right)$, for $1 \le c \le K$ and $1 \le j \le J$.

4. Let $\boldsymbol{\Sigma}_c = \mathbb{I}_J \boldsymbol{S}_c$, where $\boldsymbol{S}_c \in \mathbb{R}^{J \times J}$ is a random symmetric, positive-definite matrix,[1] for $1 \le c \le K$.

5. Sample $\boldsymbol{z}_i \sim \mathcal{N}\left(\boldsymbol{\mu}_{c_i}, \boldsymbol{\Sigma}_{c_i}\right)$, for $1 \le i \le N$.

6. Let $g_{\text{dec}}(\boldsymbol{z}) = \boldsymbol{W}_2 \text{ReLU}\left(\boldsymbol{W}_1 \text{ReLU}\left(\boldsymbol{W}_0 \boldsymbol{z} + \boldsymbol{b}_0\right) + \boldsymbol{b}_1\right) + \boldsymbol{b}_2$, where $\boldsymbol{W}_0 \in \mathbb{R}^{h \times J}$, $\boldsymbol{W}_1 \in \mathbb{R}^{h \times h}$, $\boldsymbol{W}_2 \in \mathbb{R}^{D \times h}$ and $\boldsymbol{b}_0 \in \mathbb{R}^h$, $\boldsymbol{b}_1 \in \mathbb{R}^h$, $\boldsymbol{b}_2 \in \mathbb{R}^D$ are random matrices and vectors,[2] and $h$ is the number of hidden units.

7. Let $\boldsymbol{x}_i = g_{\text{dec}}(\boldsymbol{z}_i)$, for $1 \le i \le N$.

8. Sample $\beta_{c,j} \sim \text{unif}\left(-10, 10\right)$, for $1 \le c \le K$ and $0 \le j \le J$.

9. Sample $u_i \sim \text{Weibull}\left(\text{softplus}\left(\boldsymbol{z}_i^\top \boldsymbol{\beta}_{c_i}\right), k\right)$,[3] for $1 \le i \le N$.

---

[1] Generated using `make_spd_matrix` from scikit-learn (Pedregosa et al., 2011).

[2] $\boldsymbol{W}_0, \boldsymbol{W}_1, \boldsymbol{W}_2$ are generated using `make_low_rank_matrix` from scikit-learn (Pedregosa et al., 2011).

[3] We omitted bias terms for the sake of brevity.

10. Sample $\delta_i \sim \text{Bernoulli}(1 - p_{\text{cens}})$, for $1 \leq i \leq N$.

11. Let $t_i = u_i$ if $\delta_i = 1$, and sample $t_i \sim \text{unif}(0, u_i)$ otherwise, for $1 \leq i \leq N$.

Here, similar to Section 3, $K$ is the number of clusters; $N$ is the number of data points; $J$ is the number of latent variables; $D$ is the number of features; $k$ is the shape parameter of the Weibull distribution (see Equation 1); and $p_{\text{cens}}$ denotes the probability of censoring. The key difference between clusters is in how latent variables $\boldsymbol{z}$ generate the survival time $t$: each cluster has a different weight vector $\boldsymbol{\beta}$ that determines the scale of the Weibull distribution.

In our experiments, we fix $K = 3$, $N = 60000$, $J = 16$, $D = 1000$, $k = 1$, and $p_{\text{cens}} = 0.3$. We hold out 18000 data points as a test set and repeat simulations 5 times independently. A visualisation of one simulation is shown in Figure 9. As can be seen in the $t$-SNE (van der Maaten & Hinton, 2008) plot, three clusters are not as clearly separable based on covariates alone; on the other hand, they have visibly distinct survival curves as shown in the KM plot.

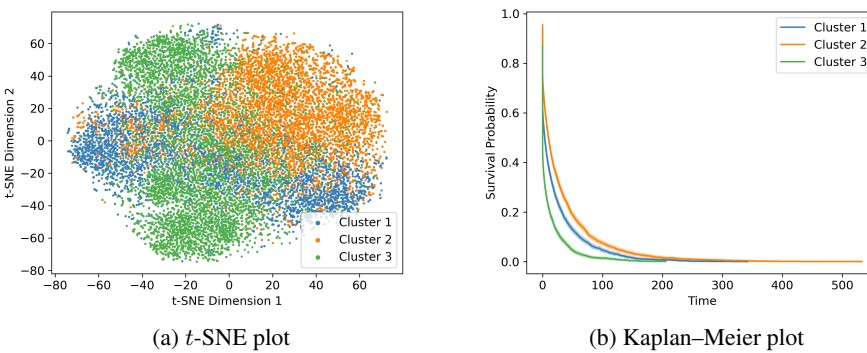

(a) $t$-SNE plot       (b) Kaplan–Meier plot

Figure 9: Visualisation of one replicate of the synthetic data. $t$-SNE plot (*on the left*) is based on the explanatory variables only. Kaplan–Meier curves (*on the right*) are plotted separately for each cluster. Different colours correspond to different clusters. As can be seen, clusters are determined by both differences in (a) covariate and (b) survival distributions.

### E.2   SURVIVAL MNIST DATA GENERATION

We generate semi-synthetic survMNIST data as described by Pölsterl (2019) (original implementation available at https://github.com/sebp/survival-cnn-estimator). The generative process can be summarised by the following steps:

1. Assign every digit in the original MNIST dataset (LeCun et al., 2010) to one of $K$ clusters (ensure that every cluster contains at least one digit)

2. Sample risk score $r_c \sim \text{unif}\left(\frac{1}{2}, 15\right)$, for $1 \leq c \leq K$.

3. Let $\lambda_c = \frac{1}{T_0} \exp(r_c)$, for $1 \leq c \leq K$, where $T_0$ is the specified mean survival time and $\frac{1}{T_0}$ is the baseline hazard.

4. Sample $a_i \sim \text{unif}(0, 1)$ and let $u_i = -\frac{\log a_i}{\lambda_{c_i}}$, for $1 \leq i \leq N$.

5. Let $q_{\text{cens}} = q_{1 - p_{\text{cens}}}(u_{1:N})$, where $q_\alpha(\cdot)$ denotes the $\alpha$-th quantile.

6. Sample $t_{\text{cens}} \sim \text{unif}\left(\min_{1 \leq i \leq N} u_i, q_{\text{cens}}\right)$.

7. Let $\delta_i = 1$ if $u_i \leq t_{\text{cens}}$, and $\delta_i = 0$ otherwise, for $1 \leq i \leq N$.

8. Let $t_i = u_i$ if $\delta_i = 1$, and $t_i = t_{\text{cens}}$ otherwise, for $1 \leq i \leq N$.

Observe that here $p_{\text{cens}}$ is only a lower bound on the probability of censoring and the fraction of censored observations can be much larger than $p_{\text{cens}}$.

Table 7: An example of an assignment of MNIST digits to clusters and risk scores under $K = 5$.

| Cluster | Digits | Risk Score |
|---------|--------|------------|
| 1 | $\{4, 8\}$ | 7.62 |
| 2 | $\{0, 3, 6\}$ | 13.12 |
| 3 | $\{1, 9\}$ | 5.17 |
| 4 | $\{2, 7\}$ | 11.60 |
| 5 | $\{5\}$ | 2.03 |

In our experiments, we fix $K = 5$ and $p_{\text{cens}} = 0.3$. We repeat simulations independently 10 times. The train-test split is defined as in the original MNIST data (60,000 vs. 10,000). Table 7 shows an example of cluster and risk score assignments in one of the simulations. The resulting dataset is visualised in Figure 10. Observe that the clusters are not well-distinguishable based on covariates alone. Also note that some clusters contain many censored data points, whereas some do not contain any, as shown in the Kaplan–Meier plot.

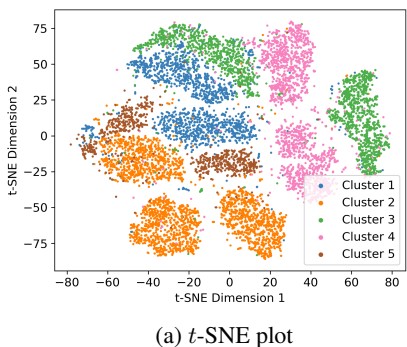

(a) $t$-SNE plot

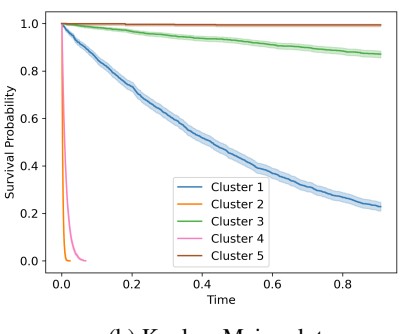

(b) Kaplan–Meier plot

Figure 10: Visualisation of one replicate of the survMNIST dataset. Generally, survMNIST tends to have more disparate survival distributions across clusters than the synthetic data (*cf.* Figure 9).

## F    EVALUATION METRICS

Below we provide details about the evaluation metrics used to compare clustering and time-to-event prediction performance of the models. For clustering, we employ the adjusted Rand index (ARI) and normalised mutual information (NMI) as implemented in the `sklearn.metrics` module of scikit-learn (Pedregosa et al., 2011). Clustering accuracy is evaluated by finding an optimal mapping between assignments and true cluster labels using the Hungarian algorithm, as implemented in `sklearn.utils.linear_assignment_`.

The concordance index (Raykar et al., 2007) evaluates how well a survival model is able to rank individuals in terms of their risk. Given observed survival times $t_i$, predicted risk scores $\eta_i$, and censoring indicators $\delta_i$, the concordance index is defined as

$$\text{CI} = \frac{\sum_{i=1}^{N} \sum_{j=1}^{N} \mathbf{1}_{t_j < t_i} \mathbf{1}_{\eta_j > \eta_i} \delta_j}{\sum_{i=1}^{N} \sum_{j=1}^{N} \mathbf{1}_{t_j < t_i} \delta_j}. \tag{13}$$

Thus, a perfect ranking achieves a CI of 1.0, whereas a random ranking is expected to have a CI of 0.5. We use the lifelines implementation of the CI (Davidson-Pilon et al., 2021).

Relative absolute error (Yu et al., 2011) evaluates predicted survival times in terms of their relative deviation from the observed survival times. In particular, given predicted survival times $\hat{t}_i$, for non-censored data points, RAE is defined as

$$\text{RAE}_{\text{nc}} = \frac{\sum_{i=1}^{N} \left| \left( \hat{t}_i - t_i \right) / \hat{t}_i \right| \delta_i}{\sum_{i=1}^{N} \delta_i}. \tag{14}$$

On censored data, similarly to Chapfuwa et al. (2020), we define the metric as follows:

$$\text{RAE}_{\text{c}} = \frac{\sum_{i=1}^{N} \left| \left( \hat{t}_i - t_i \right) / \hat{t}_i \right| (1 - \delta_i) \mathbf{1}_{\hat{t}_i \leq t_i}}{\sum_{i=1}^{N} \left( 1 - \delta_i \right)}. \tag{15}$$

Last but not least, similar to Chapfuwa et al. (2020) we use the calibration slope to evaluate the *overall* calibration of models. It is desirable for the predicted survival times to match with the empirical marginal distribution of $t$. Calibration slope indicates whether a model tends to under- or overestimate risk on average, and thus, an ideal calibration slope is 1.0.

# G   IMPLEMENTATION DETAILS

Herein, we provide implementation details for the experiments described in Section 5.

## G.1   PREPROCESSING

For all datasets, we re-scale survival times to be in $[0.001, 1.001]$. For survMNIST, we re-scale the features to be in $[0, 1]$. For tabular datasets, we re-scale the features to zero-mean and unit-variance. Categorical features are encoded using dummy variables. For Hemodialysis data, the same preprocessing routine is adopted as described by Gotta et al. (2021).

For NSCLC CT images, preprocessing consists of several steps. For each subject, slices within 15 mm from a slice with a maximal transversal tumour area are averaged to produce a single 2D image. 2D images are then cropped around the lungs and normalised. Subsequently, histogram equalisation is applied (Lehr & Capek, 1985). The images are then downscaled to a resolution of $64 \times 64$ pixels. Figure 11 provides an example of raw and preprocessed images from the Lung1 dataset. Finally, during VaDeSC, SCA, and DSM training, we augment images by randomly reducing/increasing the brightness, adding Poisson noise, flipping horizontally, blurring, zooming, stretching, and shifting both vertically and horizontally (see Figure 11). During initial experimentation, we observed that image augmentation was crucial for decorrelating clinically irrelevant scan characteristics, such as rotation or scaling, and the predicted survival outcome. Neither VaDeSC, nor DSM, achieve predictive performance comparable to radiomics-based features in the absence of augmentation.

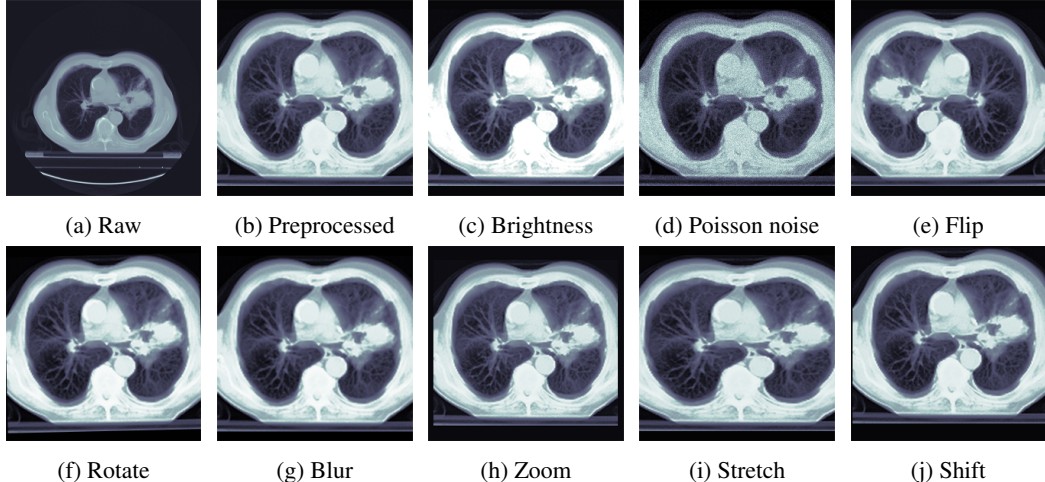

|     |     |     |     |     |
| --- | --- | --- | --- | --- |
| (a) Raw | (b) Preprocessed | (c) Brightness | (d) Poisson noise | (e) Flip |
| (f) Rotate | (g) Blur | (h) Zoom | (i) Stretch | (j) Shift |

Figure 11: A full resolution Lung1 image averaged across 11 CT slices (a) before and (b) after cropping, normalisation, and histogram equalisation. Panels (c)-(j) depict augmentations applied to images during neural network training.

For the radiomics-based Cox PH and Weibull AFT models, features were extracted from the pre-processed 2D images using the `RadiomicsFeatureExtractor` provided by the PyRadiomics library (van Griethuysen et al., 2017). Tumour segmentation maps were given as input during feature extraction, to detect the ROI.

## G.2 Models, Architectures & Hyperparameters

The following implementations of the models were used:

- **VaDeSC**: we implemented our model in TensorFlow 2.4 (Abadi et al., 2015). The code is available at `https://github.com/i6092467/vadesc`.
- $k$-**means**: we used the implementation available in the scikit-learn library (Pedregosa et al., 2011).
- **Cox PH & Weibull AFT**: we used the implementation available in the lifelines library (Davidson-Pilon et al., 2021).
- **Radiomics**: radiomic features were extracted using the PyRadiomics library (van Griethuysen et al., 2017).
- **SSC**: we re-implemented the model in Python based on the original R (R Core Team, 2020) code provided by Bair & Tibshirani (2004).
- **SCA**: we adapted the original code by Chapfuwa et al. (2020) available at `https://github.com/paidamoyo/survival_cluster_analysis`.
- **DSM**: we adapted the original code by Nagpal et al. (2021a) available at `https://github.com/autonlab/DeepSurvivalMachines`.
- **Profile regression**: we used the original implementation available in the R package PReMiuM (Liverani et al., 2015).

Throughout our experiments, we use several different encoder and decoder architectures described in Tables 8, 9, and 10. As mentioned before, the encoder architectures and numbers of latent dimensions were kept fixed across all neural-network-based techniques, namely VaDeSC, SCA, and DSM, for the sake of fair comparison.

**Hyperparameters** VaDeSC hyperparameters across all datasets are reported in Table 11. We use $L = 1$ Monte Carlo samples for the SGVB estimator. In most datasets, we do not pretrain autoencoder weights. Generally, we observe little variability in performance when adjusting the number of latent dimensions, shape of the Weibull distribution or mini-batch size. For the SCA model, we tuned the concentration parameter of the Dirichlet process, mini-batch size, learning rate, and the number of training epochs. For the sake of fair comparison, we truncate the Dirichlet process at the true number of clusters, when known. Similarly, for the DSM, the mini-batch size, learning rate, and the number of training epochs were tuned. To make the comparison with VaDeSC fair, we fixed the discount parameter for the censoring loss at $1.0$, since VaDeSC does not discount censored observations in the likelihood, and used the same number of mixture components as for the VaDeSC.

**Pretraining** VAE-based models (Dilokthanakul et al., 2016; Jiang et al., 2017; Kingma & Welling, 2014) tend to converge to undesirable local minima or saddle points, due to high reconstruction error at the beginning of the training. To prevent this, the encoder and decoder networks are usually pretrained using an autoencoder loss. In the most of our experiments, we did not observe such behaviour. Hence, we did not pretrain the model, except for the Hemodialysis dataset. The latter showed better performance if pretrained for only 1 epoch. After the encoder-decoder network has been pretrained, we projected the training samples into the latent space and fitted a Gaussian mixture model to initialise the parameters $\boldsymbol{\pi}$, $\boldsymbol{\mu}$, $\boldsymbol{\Sigma}$.

Table 8: Encoder and decoder architectures used for synthetic, survMNIST, and HGG data. `tfkl` stands for `tensorflow.keras.layers`. `encoded_size` corresponds to the number of latent dimensions, $J$; and `inp_shape` is the number of input dimesions, $D$. For survMNIST, the activation of the last decoder layer is set to `'sigmoid'`.

| | Encoder |
|---|---|
| **1** | `tfkl.Dense(500, activation='relu')` |
| **2** | `tfkl.Dense(500, activation='relu')` |
| **3** | `tfkl.Dense(2000, activation='relu')` |
| **4** | `mu = tfkl.Dense(encoded_size, activation=None);` `sigma = tfkl.Dense(encoded_size, activation=None)` |

| | Decoder |
|---|---|
| **1** | `tfkl.Dense(2000, activation='relu')` |
| **2** | `tfkl.Dense(500, activation='relu')` |
| **3** | `tfkl.Dense(500, activation='relu')` |
| **4** | `tfkl.Dense(inp_shape)` |

Table 9: Encoder and decoder architectures used for SUPPORT, FLChain, and Hemodialysis data.

| | Encoder |
|---|---|
| **1** | `tfkl.Dense(50, activation='relu')` |
| **2** | `tfkl.Dense(100, activation='relu')` |
| **3** | `mu = tfkl.Dense(encoded_size, activation=None);` `sigma = tfkl.Dense(encoded_size, activation=None)` |

| | Decoder |
|---|---|
| **1** | `tfkl.Dense(100, activation'relu')` |
| **2** | `tfkl.Dense(50, activation='relu')` |
| **3** | `tfkl.Dense(inp_shape)` |

Table 10: Encoder and decoder architectures used for NSCLC data, based on VGG (de-)convolutional blocks (Simonyan & Zisserman, 2015).

| | **Encoder** |
|---|---|
| | `// VGG convolutional block` |
| **1** | `tfkl.Conv2D(32, 3, activation='relu')` |
| **2** | `tfkl.Conv2D(32, 3, activation='relu')` |
| **3** | `tfkl.MaxPooling2D((2, 2))` |
| | `// VGG convolutional block` |
| **4** | `tfkl.Conv2D(64, 3, activation='relu')` |
| **5** | `tfkl.Conv2D(64, 3, activation='relu')` |
| **6** | `tfkl.MaxPooling2D((2, 2))` |
| **7** | `tfkl.Flatten()` |
| **8** | `mu = tfkl.Dense(encoded_size, activation=None);` `sigma = tfkl.Dense(encoded_size, activation=None)` |

| | **Decoder** |
|---|---|
| **1** | `tfkl.Dense(10816, activation=None)` |
| **2** | `tfkl.Reshape(target_shape=(13, 13, 64))` |
| | `// VGG deconvolutional block` |
| **3** | `tfkl.UpSampling2D((2, 2))` |
| **4** | `tfkl.Conv2DTranspose(64, 3, activation='relu')` |
| **5** | `tfkl.Conv2DTranspose(64, 3, activation='relu')` |
| | `// VGG deconvolutional block` |
| **6** | `tfkl.UpSampling2D((2, 2))` |
| **7** | `tfkl.Conv2DTranspose(32, 3, activation='relu')` |
| **8** | `tfkl.Conv2DTranspose(32, 3, activation='relu')` |
| **9** | `tfkl.Conv2DTranspose(1, 3, activation=None)` |

Table 11: Hyperparameters used for training the VaDeSC across all datasets. Herein, $J$ denotes the number of latent dimensions; $K$ is the number of clusters; and $k$ is the shape parameter of the Weibull distribution. MSE stands for mean squared error; BCE – for binary cross entropy.

| **Dataset** | $J$ | $K$ | $k$ | **Mini-batch Size** | **Learning Rate** | **# epochs** | **Rec. Loss** | **Pretrain # epochs** |
|---|---|---|---|---|---|---|---|---|
| Synthetic | 16 | 3 | 1.0 | 256 | 1e-3 | 1,000 | MSE | 0 |
| survMNIST | 16 | 5 | 1.0 | 256 | 1e-3 | 300 | BCE | 0 |
| SUPPORT | 16 | 4 | 2.0 | 256 | 1e-3 | 300 | MSE | 0 |
| FLChain | 4 | 2 | 0.5 | 256 | 1e-3 | 300 | MSE | 0 |
| HGG | 16 | 3 | 2.0 | 256 | 1e-3 | 300 | MSE | 0 |
| Hemodialysis | 16 | 2 | 3.0 | 256 | 2e-3 | 500 | MSE | 1 |
| NSCLC | 32 | 4 | 1.0 | 64 | 1e-3 | 1,500 | MSE | 0 |

# H  FURTHER RESULTS

## H.1  TRAINING SET CLUSTERING PERFORMANCE

Table 12: Training set clustering performance on synthetic and survMNIST data. Averages and standard deviations are reported across 5 and 10 independent simulations, respectively. Best results are shown in **bold**, second best – in *italic*. The results are similar to the test-set performance reported in Table 3.

| Dataset | Method | ACC | NMI | ARI | CI |
|---|---|---|---|---|---|
| Synthetic | $k$-means | 0.44±0.04 | 0.06±0.04 | 0.05±0.03 | — |
| | Cox PH | — | — | — | 0.78±0.02 |
| | SSC | 0.44±0.02 | 0.08±0.04 | 0.06±0.02 | — |
| | SCA | 0.45±0.09 | 0.05±0.05 | 0.05±0.05 | *0.83±0.02* |
| | DSM | 0.39±0.04 | 0.02±0.01 | 0.02±0.02 | 0.76±0.01 |
| | VAE + Weibull | 0.46±0.06 | 0.09±0.04 | 0.09±0.04 | 0.72±0.02 |
| | VaDE | 0.73±0.21 | 0.53±0.12 | 0.55±0.21 | — |
| | VaDeSC (w/o $t$) | *0.88±0.03* | *0.60±0.07* | *0.67±0.07* | **0.85±0.01** |
| | VaDeSC (ours) | **0.90±0.02** | **0.66±0.05** | **0.74±0.05** | |
| survMNIST | $k$-means | 0.48±0.06 | 0.30±0.04 | 0.21±0.04 | — |
| | Cox PH | — | — | — | 0.77±0.05 |
| | SSC | 0.48±0.06 | 0.30±0.04 | 0.21±0.04 | — |
| | SCA | 0.56±0.09 | 0.47±0.07 | 0.34±0.11 | **0.83±0.05** |
| | DSM | 0.52±0.10 | 0.38±0.17 | 0.28±0.15 | *0.82±0.05* |
| | VAE + Weibull | 0.48±0.06 | 0.31±0.06 | 0.24±0.06 | 0.80±0.08 |
| | VaDE | 0.47±0.07 | 0.38±0.08 | 0.24±0.07 | — |
| | VaDeSC (w/o $t$) | *0.57±0.10* | *0.52±0.10* | *0.37±0.10* | **0.83±0.06** |
| | VaDeSC (ours) | **0.58±0.10** | **0.54±0.12** | **0.39±0.11** | |

## H.2  QUALITATIVE RESULTS: SYNTHETIC DATA

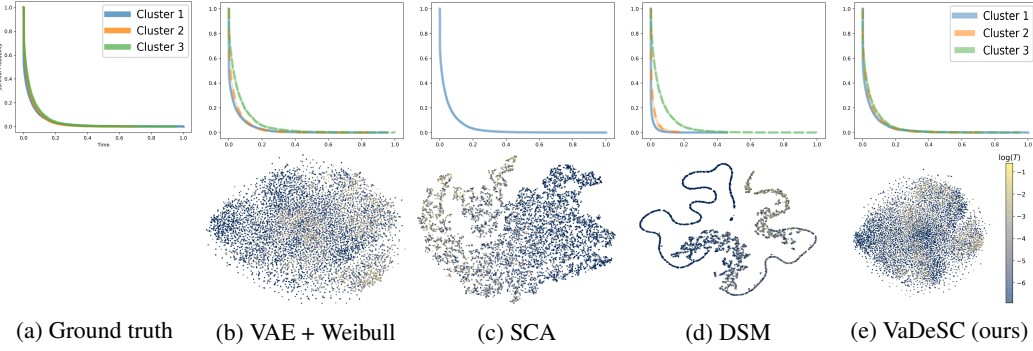

(a) Ground truth    (b) VAE + Weibull    (c) SCA    (d) DSM    (e) VaDeSC (ours)

Figure 12: Cluster-specific Kaplan–Meier (KM) curves (*top*) and $t$-SNE visualisation of latent representations (*bottom*), coloured according to survival times (yellow and blue correspond to higher and lower survival times, respectively), learnt by different models (b-e) from one replicate of the synthetic dataset. Panel (a) shows KM curves of the ground truth clusters. Plots were generated using 10,000 data points randomly sampled from the training set, similar results were observed on the test set.

## H.3  COMPARISON WITH PROFILE REGRESSION

Herein, we provide additional experimental results to compare VaDeSC with the profile regression (PR) for survival data (Liverani et al., 2020). The comparison is performed on a subsample of the original synthetic dataset (see Appendix E.1) consisting of $N = 5000$ data points. Prior to performing profile regression, we reduce the dimensionality of the dataset by retaining only the first 100 principal components, which preserve, approximately, 90% of the variance. We fit a profile

survival regression model with the cluster-specific shape parameter of the Weibull distribution by running 10000 iterations of the MCMC in the burn-in period and 2000 iterations after the burn-in. To facilitate fair comparison, we set the initial number of clusters to the ground truth $K = 3$. For VaDeSC, the same hyperparameters are used as reported in Table 11.

Table 13 reports clustering performance on the test set of 1500 data points. VaDeSC, both given survival times at prediction and without them, outperforms profile regression by a margin. Nevertheless, PR offers a noticeable improvement over $k$-means clustering and even more sophisticated SCA and DSM (*cf.* Table 3). While in some instances, PR manages to identify some cluster structure, in others, it assigns all data points to a single cluster, hence, a large standard deviation across simulations. It is also interesting to see that a reduction in the dataset size has led to a significant decrease in the performance of VaDeSC. We attribute this to a highly nonlinear structure of this dataset.

Table 13: Test set clustering performance on a subsample of the synthetic dataset. Averages and standard deviations are reported across 5 independent simulations. VaDeSC outperforms the PR baseline by a large margin.

| Method | ACC | NMI | ARI |
|---|---|---|---|
| $k$-means | 0.44±0.02 | 0.07±0.04 | 0.05±0.02 |
| PR | 0.47±0.13 | 0.15±0.17 | 0.15±0.17 |
| VaDeSC (w/o $t$) | *0.73±0.08* | *0.34±0.11* | *0.38±0.13* |
| VaDeSC (ours) | **0.77±0.09** | **0.41±0.13** | **0.45±0.16** |

This experiment has clear shortcomings: due to computational limitations, we are only able to fit a profile regression model on a transformed version of the dataset with fewer dimensions which might not reflect the full capacity of the PR model. Nevertheless, we believe that poor scalability of the MCMC, both in the number of covariates and in the number of training data points, warrant a more computationally efficient approach.

## H.4 VARYING THE FRACTION OF CENSORED DATA POINTS

Herein we investigate the behaviour of the proposed approach under different fractions of censored observations. We perform a controlled experiment on the synthetic dataset (see Appendix E.1) where we vary the percentage of censored data while keeping all other simulation parameters fixed. We evaluate the proposed algorithm, VaDeSC, and several baselines (VaDE, DSM, and Weibull AFT) under 10-90% of censored observations and compare their clustering and time-to-event predictive performance.

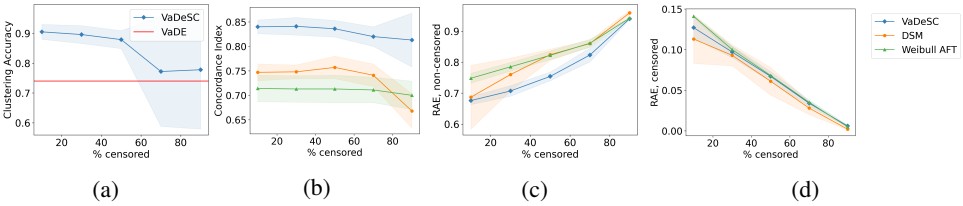

Figure 13: Performance of the **VaDeSC** on the synthetic data (see Appendix E.1) at varying percentages of censored data points (10-90%): (a) clustering accuracy, (b) concordance index, (c) relative absolute error (RAE) evaluated on non-censored and (d) censored data. Averages and standard deviations are reported across 5 independent simulations, evaluation was performed on test data. For reference, we report performance of the unsupervised clustering model **VaDE** and of **DSM** and **Weibull AFT** models for survival analysis.

Figure 13 shows the results of the experiment. VaDeSC stably achieves clustering accuracy of, approximately, 90% for 10, 30, and 50% of censored observations. However, for 70 and 90%, its accuracy drops considerably, and variance across simulations increases. Nevertheless, even at 90% of censored data points, on average, VaDeSC is still more accurate and stable than the completely

unsupervised VaDE. For time-to-event predictions, all models behave as expected: with an increase in the percentage of censored observations, test-set CI decreases, $RAE_{nc}$ increases, while $RAE_c$ decreases. Overall, VaDeSC, DSM, and Weibull AFT appear to behave and scale very similarly w.r.t. changes in censoring. In the future, it would be interesting to perform a similar experiment on a real-world dataset with a moderate percentage of censored observations, *e.g.* SUPPORT.

## H.5 VARYING THE NUMBER OF COMPONENTS IN VADESC

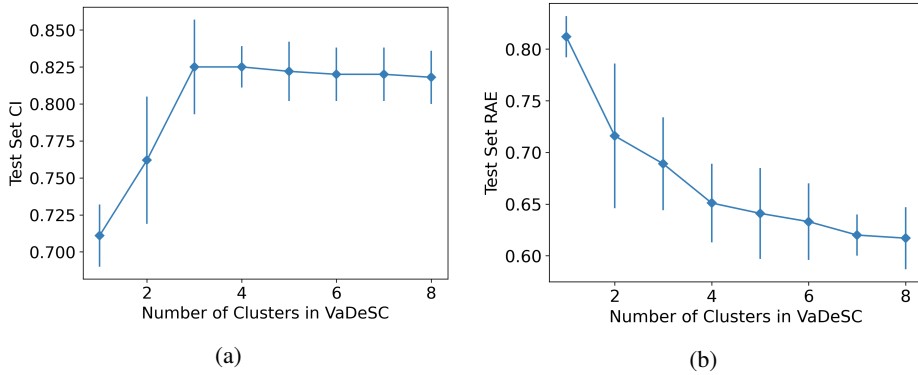

(a)                              (b)

Figure 14: Average test set (a) CI and (b) RAE on synthetic data with $K = 3$ clusters achieved by VaDeSC models with different numbers of mixture components. Error bars correspond to standard deviations across 5 independent simulations. The CI plot (*left*) exhibits a pronounced 'elbow' at the true number of clusters.

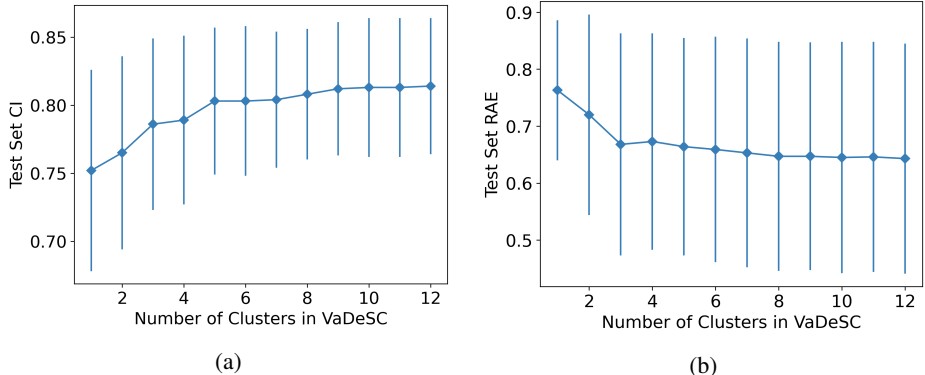

(a)                              (b)

Figure 15: Average test set (a) CI and (b) RAE on survMNIST with $K = 5$ clusters achieved by VaDeSC models with different numbers of mixture components. Error bars correspond to standard deviations across 10 independent simulations. The CI plot (*left*) appears to saturate at the true number of clusters, although the 'elbow' is not quite as sharp as for the synthetic data (*cf.* Figure 14).

## H.6 RECONSTRUCTIONS & GENERATED SURVMNIST SAMPLES

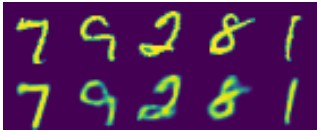

Figure 16: Reconstructions of survMNIST digits. Original digits are shown in the top row, their reconstructions by VaDeSC — in the bottom row.

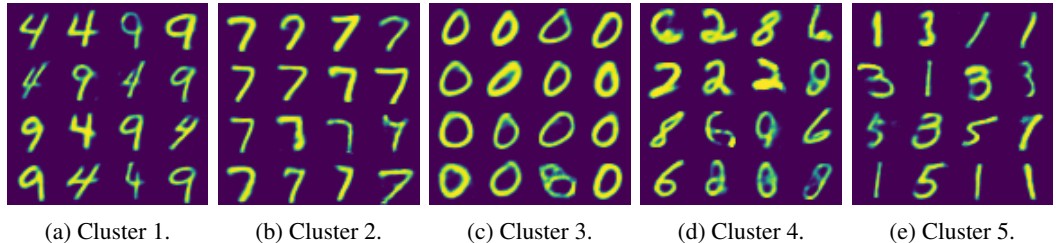

| (a) Cluster 1. | (b) Cluster 2. | (c) Cluster 3. | (d) Cluster 4. | (e) Cluster 5. |

Figure 17: survMNIST samples generated by VaDeSC. In this simulation, the true clustering is given by $\{\{7\}, \{0, 9\}, \{2, 4, 6\}, \{8\}, \{1, 3, 5\}\}$. Cluster 1 appears to align with $\{0, 9\}$; cluster 2 — with $\{7\}$; cluster 4 — with $\{2, 4, 6\}$; cluster 5 — with $\{1, 3, 5\}$.

## H.7 TIME-TO-EVENT PREDICTION: FLCHAIN

Table 14: Time-to-event test set performance on FLChain. Averages and standard deviations are reported across 5 train-test splits. All of the methods perform comparably w.r.t. CI. SCA achieves better RAE and calibration.

| Method | CI | $\text{RAE}_{nc}$ | $\text{RAE}_c$ | CAL |
|---|---|---|---|---|
| Cox PH | **0.80±0.01** | — | — | — |
| Weibull AFT | **0.80±0.01** | *0.72±0.01* | *0.02±0.00* | *2.18±0.07* |
| SCA | 0.78±0.02 | **0.69±0.08** | *0.05±0.05* | **1.33±0.24** |
| DSM | *0.79±0.01* | 0.76±0.05 | **0.02±0.01** | 2.35±0.66 |
| VAE + Weibull | **0.80±0.01** | 0.76±0.01 | **0.02±0.00** | 2.55±0.07 |
| VaDeSC (ours) | **0.80±0.01** | 0.76±0.01 | **0.02±0.00** | 2.52±0.08 |

## H.8 QUALITATIVE RESULTS: HEMODIALYSIS DATA

In addition to the results reported in Section 5.2, we have investigated qualitatively the clustering obtained by VaDeSC when applied to the Hemodialysis dataset. The true structure of this data is unknown, however several studies have suggested a stratification of patients according to age and dialysis doses (Gotta et al., 2021). We compute the optimal number of clusters using grid-search by measuring the time-to-event prediction performance. With two clusters our model achieved the best results. In Figure 18(a), we plot the $t$-SNE decomposition of the latent embeddings, $z$ (see Figure 2), obtained by VaDeSC. Observe that the patients clustered together tend to be close in the latent space, following the mixture of Gaussian prior. We also plot Kaplan–Meier curves for each patient subgroup according to the predicted cluster in Figure 18(b). Interestingly, the two resulting curves differ substantially from each other. The patients in cluster 2 (■) show a higher survival probability over time compared to those in cluster 1 (■).

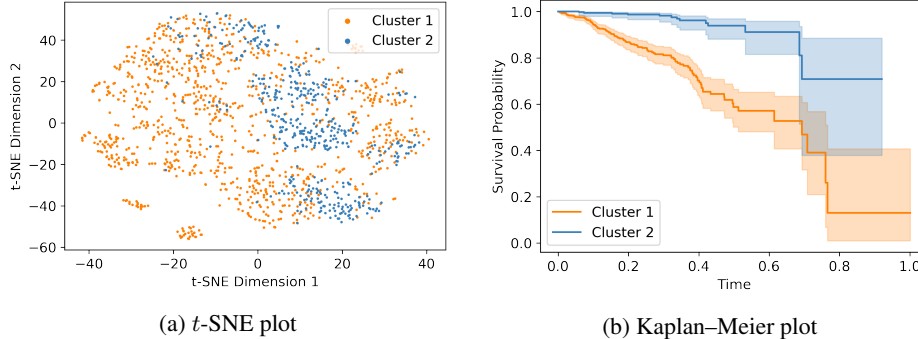

(a) $t$-SNE plot

(b) Kaplan–Meier plot

Figure 18: Visualisation of (a) the $t$-SNE decomposition of the Hemodialysis data in the embedded space and (b) the Kaplan–Meier curves for the two clusters discovered by VaDeSC. The two clusters have substantially different survival distributions.

We now characterise each subgroup of patients by identifying the most influential covariates in determining the cluster assignment by VaDeSC. In particular, we apply a state-of-the-art method based on Shapley values (Shapley, 1953), namely the TreeExplainer by Lundberg et al. (2020), to explain an XGBoost classifier (Chen & Guestrin, 2016) trained to predict the VaDeSC cluster labels from patient characteristics (see Appendix I). To mimic the VaDeSC clustering model, we included survival times as an additional input to the classifier. In Figure 20, we present the most important covariates together with their cluster-wise distributions. It is important to note that the survival time was also ranked among the most important, but we did not include it into Figure 20. Serum albumin level, previously identified as an important survival

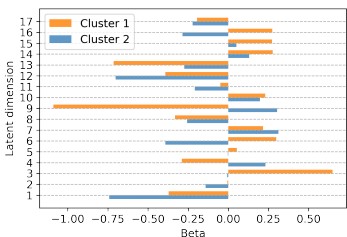

Figure 19: Cluster-wise $\boldsymbol{\beta}$ parameters of the Weibull distributions discovered on the Hemodialysis dataset.

indicator for patients receiving hemodialysis (Foley et al., 1996; Amaral et al., 2008), emerges as the most important predictor. Additionally, dialysis dose in terms of spKt/V and fluid balance in terms of ultrafiltration (UF) rate prove to be crucial in classifying the data into high-risk (cluster 1, ■) and low-risk (cluster 2, ■) groups. By analysing the cluster-wise distributions, we notice that the high-risk cluster is characterised by lower values of albumin, dialysis dose, and ultrafiltration, in agreement with previous studies (Gotta et al., 2020; 2021; Movilli et al., 2007).

**Cluster-Specific Survival Models**  Ultimately, our model should discover subgroups of patients characterised not only by their risk but, more importantly, by the relationships between the covariates and survival outcomes. In other words, the cluster-specific parameters of the Weibull distribution should ideally vary across clusters, highlighting different associations between the covariates. We demonstrate this in Figure 19 by plotting the cluster-specific survival parameters, $\boldsymbol{\beta}_c$ (see Equation 1), trained using the Hemodialysis data. We observe a clear difference between the two clusters, with several latent dimensions described by both positive and negative survival parameters depending on the considered cluster.

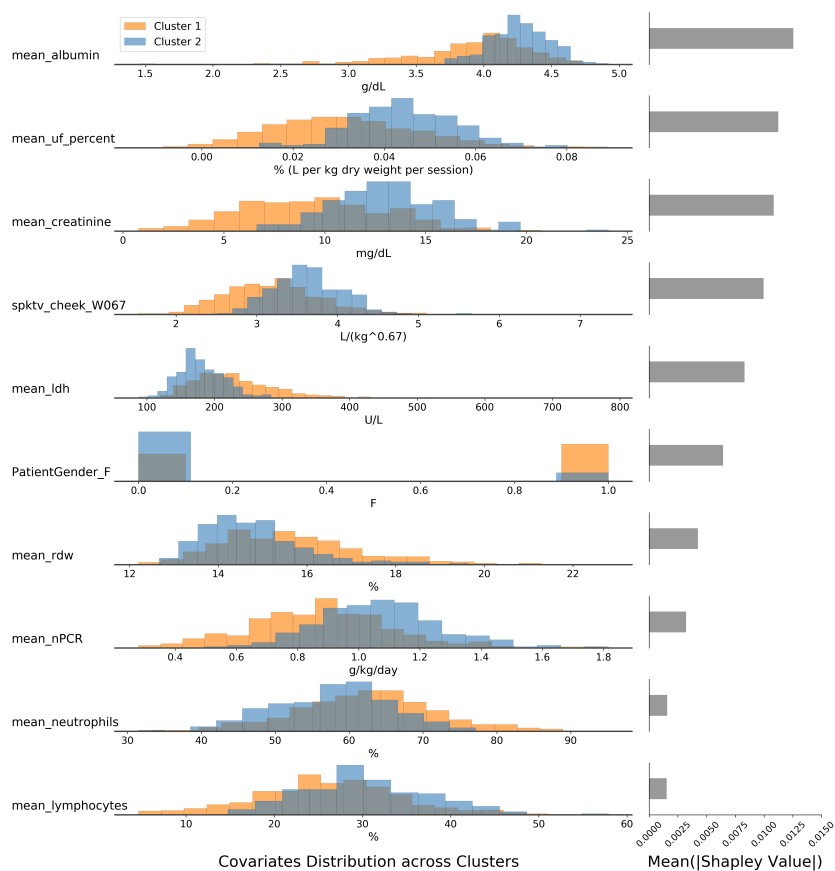

Figure 20: The most important predictors for VaDeSC cluster assignments, according to the average Shapley values (see Appendix I). The survival time was not included into this plot, although it was ranked among the most important features. The two clusters are characterised by disparate distributions of the considered clinical variables, where the high-risk cluster shows lower values of albumin, dialysis dose, and ultrafiltration in agreement with previous studies.

## H.9 QUALITATIVE RESULTS: NSCLC

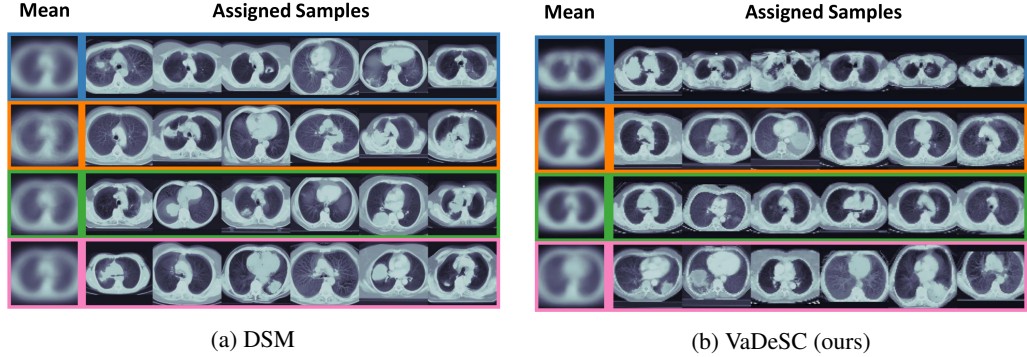

(a) DSM

(b) VaDeSC (ours)

Figure 21: A random selection of CT images assigned by (a) DSM and (b) VaDeSC to each cluster (denoted by colour) with the corresponding centroid images, computed across all samples. The colours correspond to the same clusters as in Figure 5. Clusters discovered by VaDeSC are strongly correlated with the tumour location; DSM does not appear to discover such association.

**Mean**                                                            **Generated Samples**

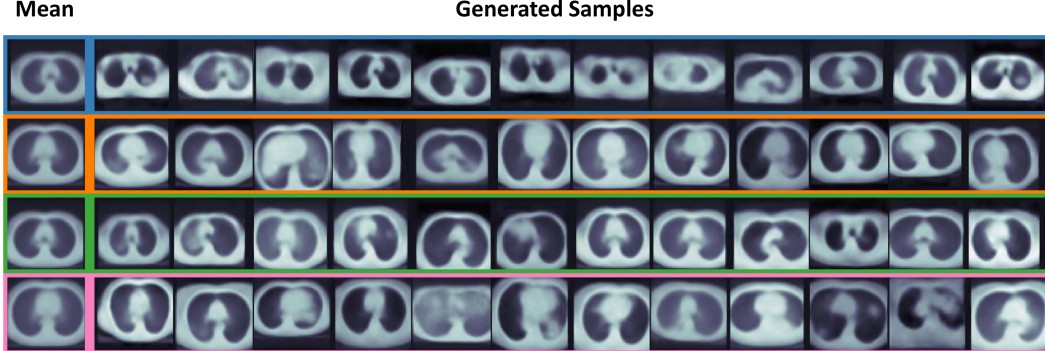

Figure 22: An extended version of Figure 6. CT images generated by (*i*) sampling latent representations from the Gaussian mixture learnt by VaDeSC and (*ii*) decoding the representations using the decoder network. The colours correspond to the same clusters as in Figure 5.

### H.10 CLUSTERING STABILITY: NSCLC

The notion of stability has been widely explored as a validation technique for clustering algorithms (Ben-Hur et al., 2002; Lange et al., 2004). The intuition is that a '*good*' algorithm should discover consistent clustering structures across repeated random simulations or replicates. However, unsupervised learning algorithms based on deep neural networks have been shown to be highly unstable on real-world and complex data (Jiang et al., 2017). Herein we test the stability of VaDeSC and compare it to a competitive baseline, DSM, on the NSCLC dataset. This dataset is extremely complex due to both the high-dimensionality and the limited number of samples. We run both methods on 20 independent train-test splits (with random seeds) and we observe the differences in the learnt clustering structures.

First of all, note that the number of discovered clusters varies across experiments. Even though the number of mixture components in survival distribution is defined *a priori*, some clusters collapse, *i.e.* they do not contain any samples from both the training and test set. We measure this source of instability by computing the percentage of experiments that resulted in at least one collapsed cluster in Table 15. We observe that only 5% of VaDeSC's runs resulted in collapsed clusters, compared to 45% for DSM. Thus, VaDeSC is relatively stable w.r.t. the number of discovered clusters.

We also test the variability in the survival information encoded in each cluster, also shown by the Kaplan–Meier curves in Figure 5. For each experiment, we compute the median survival time of the samples assigned by the algorithm to each cluster. In Table 15 we report the average and standard deviation across the experiments of the cluster-specific median survival time. To have a meaningful average, we sort clusters by the median survival time, highest to lowest. As discussed in Section 5.3, DSM tends to discover clusters with more disparate survival curves than VaDeSC, which is reflected by the averages reported. Here, we instead focus on the standard deviation. Overall, VaDeSC shows lower standard deviations than DSM. VaDeSC appears to be stable w.r.t. the survival information captured by each cluster.

Table 15: The average and the standard deviation of the cluster-specific median survival time computed across 20 independent experiments. For each experiment we re-train the model on a different train-test split. The survival time has been scaled between 0 and 1. We also report the percentage of experiments that resulted in one or more collapsed clusters.

| Method | Collapsed Clusters, % | Average Median Survival Time $\pm$ SD | | | |
|---|---|---|---|---|---|
| | | Cluster 1 | Cluster 2 | Cluster 3 | Cluster 4 |
| DSM | 45 | $0.296 \pm 0.087$ | $0.240 \pm 0.083$ | $0.138 \pm 0.055$ | $0.093 \pm \mathbf{0.032}$ |
| VaDeSC | **5** | $0.220 \pm \mathbf{0.044}$ | $0.157 \pm \mathbf{0.016}$ | $0.139 \pm \mathbf{0.016}$ | $0.086 \pm 0.038$ |

Finally, we test whether the association between tumour locations and clusters is also consistent across experiments. We first compute the means of the mixture of Gaussians learnt by VaDeSC for each experiment, *i.e.* train-test split. As these vectors live in the latent space of VaDeSC, we

decode them, using the decoder network, to project them into the input space of the CT images. Resulting images are shown in Figure 23, where each column corresponds to a different seed of the train-test split. As before, we sort the clusters according to the median survival times such that the upper row corresponds to patients with higher overall survival probability. It is evident that in most experiments, the upper row corresponds to the upper section of the lungs, whereas the bottom row — to the lower section of the lungs. Thus, VaDeSC discovers clusters consistently associated with tumour location.

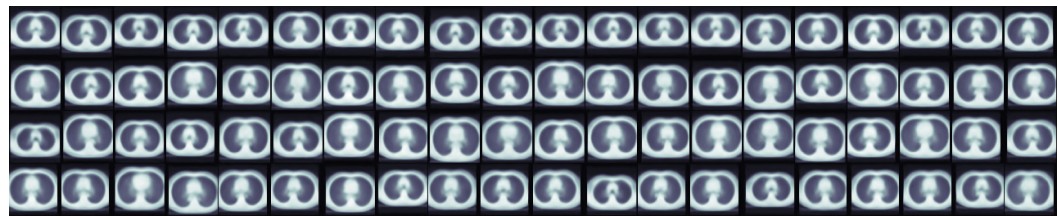

Figure 23: CT images generated by training the VaDeSC model 20 times using different train-test splits. For each trained model, we (*i*) select the learnt means of the Gaussian mixture and (*ii*) decode them using the decoder network. Each column corresponds to a different seed of the train-test split used to train the model. Each row corresponds to a different cluster. For each seed, we order the rows according to the median survival within the cluster, hence the upper rows are associated with higher survival time. We observe a clear association with tumour location, consistent across most experiments.

## I   EXPLAINING VADESC CLUSTER ASSIGNMENTS

In this appendix, we detail the *post hoc* analysis performed to enrich our case study on the real-world Hemodialysis dataset (see Appendix H.8) by explaining VaDeSC cluster assignments in terms of input covariates. Similar analysis can be performed for any survival dataset where an explanation of VaDeSC clusters in terms of salient clinical concepts is desirable.

**Background: SHAP algorithm for interpreting model predictions**   SHapley Additive exPlanation (SHAP), by Lundberg & Lee (2017), is an algorithm that aims at explaining the prediction of an instance $\boldsymbol{x}$ by computing the contribution of each feature to the prediction. This contribution is estimated in terms of Shapley regression values, an additive feature attribution method inspired by the coalitional game theory (Shapley, 1953). Additive feature attribution methods imply that the explanation model is a linear function of binary variables: $g(\boldsymbol{z}') = \phi_0 + \sum_{j=1}^{F} \phi_j z'_j$ where $g(\cdot)$ is the explanation model, $\boldsymbol{z}' \in \{0,1\}^F$ is the binary feature vector of dimension $F$, and $\phi_j \in \mathbb{R}$ is the effect of the $j$-th feature on the output $f(\boldsymbol{x})$ of the original model.

Shapley regression values estimate this effect by exploiting ideas drawn from the coalitional game theory. In particular, for the $j$-th variable the Shapley regression values are computed as

$$\phi_j = \sum_{\mathcal{S} \subseteq \mathcal{F} \setminus \{j\}} \frac{|\mathcal{S}|!(|\mathcal{F}| - |\mathcal{S}| - 1)!}{|\mathcal{F}|!} \left[ f_{\mathcal{S} \cup \{j\}} \left( \boldsymbol{x}_{\mathcal{S} \cup \{j\}} \right) - f_{\mathcal{S}}(\boldsymbol{x}_{\mathcal{S}}) \right], \tag{16}$$

where $\mathcal{F} = \{1, ..., F\}$ is the set of all predictor variables, $\boldsymbol{x}_{\mathcal{S} \cup \{j\}}$ and $\boldsymbol{x}_{\mathcal{S}}$ are input vectors from $\boldsymbol{x}$, composed of the features in the subset $\mathcal{S} \subseteq \mathcal{F}$ with and without the $j$-th feature, respectively, and $f(\cdot)$ is the original model fitted on the set of predictor variables defined by the subscript.

The feature importances estimated by SHAP are the Shapley values of a conditional expectation function of the original model (Lundberg et al., 2020). With SHAP, global interpretations are consistent with the local explanations, since the Shapley values for instance $i$ are the *atomic unit* of the global interpretations. Indeed, the global importance of the $j$-th feature is computed as

$$I_j = \frac{1}{N} \sum_{i=1}^{N} \left| \phi_j^{(i)} \right|, \tag{17}$$

where $N$ is the total number of observations used to explain the model globally.

Despite the theoretical advantages of SHAP values, their practical use is hindered by the complexity of estimating $\mathbb{E}\left[f(\boldsymbol{x})|\boldsymbol{x}_\mathcal{S}\right]$ efficiently. TreeExplainer (Lundberg et al., 2020) is a variant of SHAP algorithm that computes the classical Shapley values from the game theory specifically for trees and tree ensembles reducing the complexity from exponential to polynomial time.

**Preprocessing** With the goal of explaining VaDeSC cluster assignments in terms of input covariates, we preselect features manually, to make our explanations robust and relevant from a clinical perspective. Guided by the medical expertise, we exclude highly correlated or redundant features. While the VaDeSC can handle the entire raw dataset meaningfully, the presence of highly correlated features might affect their relevance. Since $p(c|\boldsymbol{z}, t)$ in VaDeSC depends on survival time (see Section 3), we included the survival times as an additional input to the classifier. Note that for this *post hoc* analysis we used training data only, as we were interested in explaining what our model *learnt* from the raw data to perform clustering.

**Explanations** To characterise patient subgroups identified by VaDeSC, one needs to identify the features in the input space that maximise cluster separability in the embedding space. Note that a classifier trained to recognise cluster labels from the raw covariates is inherently seeking those dimensions of maximal separability in the input. Therefore, we first fitted a classifier model to extract the *explanations* in terms of the input features most relevant for the prediction of cluster labels.

In particular, we trained an eXtreme Gradient Boosting (XGBoost) binary classifier (Chen & Guestrin, 2016), to predict patient labels $c \in \{1, 2\}$ assigned by VaDeSC. We choose XGBoost since it has high accuracy in the presence of nonlinear relationships, naturally captures feature interactions, suffers from low bias, and supports fast exact computation of Shapley values by TreeExplainer (Lundberg et al., 2020). XGBoost was trained for 5,000 boosting rounds with a maximum tree depth of 6, a learning rate equal to 0.01, a subsampling rate of 0.5, and early stopping after 200 rounds without an improvement on the validation set (set to 10% of the training data). After training the classifier, we used the TreeExplainer to compute SHAP values and identify the most determinant features for each cluster label, as in Equation 17. To profile the clusters precisely and avoid the noise induced by weakly assigned observations, we computed SHAP values on a sample of *cluster prototypes*, *i.e.* observations with $\max_{\nu \in \{1,2\}} p(c = \nu|\boldsymbol{z}, t) \geq 0.75$. We retained at most 500 prototypes from each cluster. Due to the cluster imbalance, that resulted in 500 observations from cluster 1 and 212 – from cluster 2.

