# OpenReview forum: "A Deep Variational Approach to Clustering Survival Data"
_ICLR.cc/2022/Conference — ICLR 2022 Poster_

### Official Review · Reviewer_VEke · 2021-11-01

**Correctness:** 3
**Technical Novelty And Significance:** 2
**Empirical Novelty And Significance:** 3
**Recommendation:** 6
**Confidence:** 4

**Main Review:**

The paper is well written and easy to follow.

Nonetheless, Figure 4 is hard to read and interpret, it would be interesting to do this analysis on a dataset with a smaller number of clusters. Further description of the clustering evaluation would also be valuable: in figure 4, is the latent representation obtained on the test split only? It seems the authors use a Monte Carlo cross-validation, it would be beneficial to add this point in the experiment section.
Finally, in Figure 5, were the number of clusters and their shapes consistent over multiple iterations? An experiments measuring this stability would be a good addition.

Finally, it would be necessary to further articulate the technical contributions of the paper, as the proposed model is similar to [1] in which they also propose a variational approach to obtain a better embedding that influences the baseline distributions. The main difference resides in the baseline distributions which are parametric in [1]. The proposed appraoch merges [1] and [2] and extends it to a fully generative model. This should be clearly stated and a comparison with the distribution obtained in [1] would be a valuable addition.

[1] Nagpal, C., Yadlowsky, S., Rostamzadeh, N. and Heller, K., 2021. Deep Cox mixtures for survival regression.
[2] Nagpal, C., Li, X.R. and Dubrawski, A., 2021. Deep survival machines: Fully parametric survival regression and representation learning for censored data with competing risks.

**Summary Of The Paper:**

This work tackles the problem of clustering in the context of survival data using a generative model. A variational autoencoder is used for modelling the data, while the latent representation is leveraged to model the survival outcome conditionned on the assigned cluster following a Weibull distribution. This approach allows to leverage both the survival outcome and the covariates for clustering. Moreover, the latent state associated to each cluster is then used for interpretability of the observed cluster.
Synthetic and real world datasets demonstrate the competitiveness of the method on both discriminative and clustering performances.

**Summary Of The Review:**

An interesting work, well-described and with in-depth experiments. However, the technical contribution has to be described in greater details.

---

> ### Author Response · Authors · 2021-11-13
> **Response to Reviewer VEke**
>
> We thank the reviewer for the constructive criticism. We made several revisions in the manuscript which are briefly summarised in the official comment to all of the reviewers. Below are our point-by-point responses to your concerns.
>
> > Figure 4 is hard to read and interpret.
>
> We have updated the figure in the revised manuscript to make KM curves easier to discern. The analysis with fewer clusters ($K=3$) was performed for the synthetic dataset, see Figure 12 in Appendix H.2.
>
> > In figure 4, is the latent representation obtained on the test split only?
>
> All plots in Figure 4 were generated from 10,000 data points randomly sampled from the training set. We have observed similar results on the test data. We have amended this detail to the figure’s caption.
>
> > It seems the authors use a Monte Carlo cross-validation, it would be beneficial to add this point in the experiment section.
>
> For real-world datasets, we indeed use the Monte Carlo CV procedure. For synthetic datasets, models are evaluated on several independent simulations, i.e. the dataset is replicated multiple times. We now mention these details explicitly in Section 4 of the revised manuscript.
>
> > In Figure 5, were the number of clusters and their shapes consistent over multiple iterations? An experiment measuring this stability would be a good addition.
>
> We thank the reviewer for the insightful suggestion. We indeed believe that clustering stability is crucial in medical applications. Hence, we followed the given advice and performed various experiments on the NSCLC dataset to compare the stability of the clustering results on both the VaDeSC and DSM algorithms (see Appendix H.10). In particular, we ran 20 independent experiments (with random train-test split) and observed the differences in the clustering results for both methods. From the results shown in Table 15 we observe that VaDeSC is more stable than the baseline on both the number of clusters and the cluster-specific median of survival times. Additionally, in Figure 23 we plot the reconstructed means of the learnt mixture of Gaussians across the different experiments. It is evident that VaDeSC consistently discovers clusters with a clear association with tumour location. For further details, we refer to Appendix H.10.
>
> > The proposed approach merges [1] and [2] and extends it to a fully generative model. This should be clearly stated and a comparison with the distribution obtained in [1] would be a valuable addition.
>
> We thank the reviewer for raising this point. To the best of our knowledge, the work in [1] has been officially published in the proceedings on October 21, 2021 (see https://proceedings.mlr.press/v149/assets/bib/citeproc.yaml). We consider this work to be concurrent to ours.
>
> We have added a more detailed comparison with the approach in [1] to Section 2 of the revised manuscript. Moreover, we now provide a more detailed discussion in Appendix C of the revised paper.
>
> To summarise, we believe that the proposed model does not just merge [1] and [2]. DSM [2] is not a generative model w.r.t. covariates $\boldsymbol{x}$ and has no decoder arm. DCM [1], on the other hand, does not specify a generative model and its loss is derived empirically by combining the VAE loss with the likelihood of survival times. Moreover, DCM does not actually require the use of a VAE; a simple MLP encoder, according to the authors themselves, could be used as well. Formally, DCM algorithm maximises the likelihood of the survival time alone $\mathcal{L}(t)$ plus the likelihood of the covariates $\mathcal{L}(\boldsymbol{x})$. Given that the covariates are not independent of the survival time, this does not result in maximising the joint likelihood $\mathcal{L}\left(\boldsymbol{x},t\right)$.
>
>
> On the other hand, our approach is probabilistic and specifies a clear and interpretable generative process from which an ELBO of the **joint** likelihood can be derived formally. Additionally, VaDeSC enforces a more structured representation in the latent space, since it uses a Gaussian mixture prior, as opposed to the DCM, which uses a vanilla VAE.
>
> Another important technical contribution of the current work is the experiments and focus on image data: neither [1], nor [2], nor other related works on clustering and mixture modelling for survival data have experimented with medical imaging. We believe that this is a crucial step towards translating the success of contemporary ML to the biomedical domain.
>
> References:
>
> [1] Nagpal, C., Yadlowsky, S., Rostamzadeh, N. and Heller, K., 2021. Deep Cox mixtures for survival regression.
>
> [2] Nagpal, C., Li, X.R. and Dubrawski, A., 2021. Deep survival machines: Fully parametric survival regression and representation learning for censored data with competing risks.

---

### Official Review · Reviewer_Z7tL · 2021-11-02

**Correctness:** 4
**Technical Novelty And Significance:** 3
**Empirical Novelty And Significance:** 4
**Recommendation:** 8
**Confidence:** 3

**Main Review:**

Strengths:
- The literature review on related clinical survival analysis research is thoroughly summarized with clear statement on the relevance and differences with comparison to the proposed method.
- Experiment design is comprehensive and rigorous with various important evaluation metrics of clustering analysis and survival predictions, on different types of datasets (synthetic, semi-synthetic and real-world benchmark data)
- Detailed experiment assumptions, parameters, implementations, as well as results (enough Appendix references and supplementary material to reproduce the proposed work as well as baseline models)
- Meaningful discussion of method limitations and future work

Weaknesses:
- My major complaint originated from the potential fair practical utility of the proposed method in clinical applications indicated by Figure 5, the comparison of Kaplan-Meier curves between DSM and VaDeSC among identified clusters. One major purpose of the survival analysis is to guide bedside clinicians and nurses for timely intervention for better survival outcome and the clustering analysis could offer meaningful grouping of risk-stratified patients with intervention priorities. Thus a clear separation of different shapes of KL curves might be more helpful in real-time bedside practice and guidance, than the visual separation of images and allocation of tumors. In this case, DSM seems to perform better for this purpose in indicate "When to intervene" on the NSCLC dataset. However, VaDeSC, by its nature to differentiate variants, might be better utilized in diagnosis process for image data. I also wonder whether it's the same case for tabular data.
- Color visualization of Figure 4: KM curves of assigned clusters are very hard to see and compare. Yellow and blue might be better replaced with e.g. red and green for better contrast.
- Is there any justification for why FLChain results are in Appendix instead of main text? section 5.2 on page 7. Methods and datasets are not uniform across the paper e.g. table 2, 3 and 4 are of different subsets. If space is limited, is it possible to provide complete table of results comparison of all metrics, all methods and all datasets in the Appendix.
- The content of the paper is very dense to read due to the great efforts the author(s) have put. Sometimes the results lack immediate clear interpretation in the main text and referring to Appendix seems to be too frequent. It will be better if the main content could be more targeted and focused.

Additional question:
- Would you pls comment how the performance of different models vary by the %censored of the datasets?


**Summary Of The Paper:**

The paper proposed a novel variational deep survival clustering (VadeSC) method to discover underlying distribution of both the explanatory variables and censored survival times. Its main contributions are jointly modeling variables and censored survival outcomes and comprehensive comparison of proposed methods on state-of-art existing models on both synthetic and real-world datasets.

**Summary Of The Review:**

The paper which proposed a novel survival clustering algorithm is well-written with thorough literature review, clear analysis assumptions, comprehensive experiment design and results comparison with state-of-art baselines models and closely related benchmark datasets, which is a good example for dedicated research work.

---

> ### Author Response · Authors · 2021-11-13
> **Response to Reviewer Z7tL: Part 1**
>
>
> We thank the reviewer for the thorough review and constructive feedback. We made several revisions in the manuscript which are briefly summarised in the official comment to all of the reviewers. Below are our point-by-point responses to your concerns.
>
> > A clear separation of different shapes of KL curves might be more helpful in real-time bedside practice and guidance, than the visual separation of images and allocation of tumors. In this case, DSM seems to perform better for this purpose in indicate "When to intervene" on the NSCLC dataset. However, VaDeSC, by its nature to differentiate variants, might be better utilized in diagnosis process for image data.
>
> We partially agree with and acknowledge this limitation of VaDeSC pointed out by the reviewer. Indeed, if the only task of interest is patient management based purely on risk, DSM is an appropriate clustering technique. However, if we are interested in discovering patient subpopulations with cluster-specific interaction between the covariates and the survival time, VaDeSC and its generative model are more appropriate. In some cases, a joint relationship between $\boldsymbol{x}$ and $t$ could be important for effectively ‘intervening’ on subjects, e.g. see work by Coker et al. (2018). Many relevant examples come from cancer research, where researchers integrate clinical, treatment and genome-wide information (e.g. transcriptome, metabolome, somatic alterations, or multi-omic data) to identify groups of patients characterised not only by different risk profiles but also by different clinical and genomic characteristics to ultimately identify useful biomarkers to drive group-specific treatment decisions, such as in Tang et al. (2013) or Samur et al. (2020).
> To sum up, we nevertheless believe that the current paper presents an interesting alternative view of the survival clustering problem tailored more towards exploratory analysis and scientific discovery. Moreover, as the reviewer has mentioned, due to its generative nature, VaDeSC might be a better fit for image data. In the context of the NSCLC CT experiment, for example, our method discovered an association between tumour location and survival outcome, which could not be picked up by DSM and could be easily visualised by decoding cluster means.
>
> References:
>
> Coker, E., Liverani, S., Su, J.G. et al. Multi-pollutant Modeling Through Examination of Susceptible Subpopulations Using Profile Regression. Current Environmental Health Reports 5, 59–69 (2018). https://doi.org/10.1007/s40572-018-0177-0
>
>
> Tang Z, Ow GS, Thiery JP, Ivshina AV, Kuznetsov VA. Meta-analysis of transcriptome reveals let-7b as an unfavorable prognostic biomarker and predicts molecular and clinical subclasses in high-grade serous ovarian carcinoma. Int J Cancer. 2014 Jan 15;134(2):306-18. doi: 10.1002/ijc.28371
>
> Samur Mehmet Kemal, Anil Aktas Samur, Mariateresa Fulciniti, Raphael Szalat, Tessa Han, et al.. Genome-Wide Somatic Alterations in Multiple Myeloma Reveal a Superior Outcome Group. Journal of Clinical Oncology, American Society of Clinical Oncology, 2020, 38, pp.JCO2000461.

---

> > ### Author Response · Authors · 2021-11-13
> > **Response to Reviewer Z7tL: Part 2**
> >
> > > I also wonder whether it's the same case for tabular data.
> >
> > The whole discussion above applies to the tabular data as well.
> >
> > > Color visualization of Figure 4
> >
> > We thank the reviewer for the feedback. We have updated Figures 4 and 12 (in Appendix H.2) in the revised manuscript to make KM curves easier to compare. Note, that for SCA and DSM, the visualisation has changed, since their original code does not fix seed for TensorFlow and PyTorch. Nevertheless, the takeaway message stays the same.
> >
> > > Yellow and blue might be better replaced with e.g. red and green for better contrast.
> >
> > We have chosen these colours to be discernible by colourblind readers (red-green colourblindness is the commonest type) and be visible when printed in grayscale. We will look for a better gradient for the camera-ready version of the manuscript.
> >
> > > Is there any justification for why FLChain results are in Appendix instead of main text?
> >
> > The FLChain results were placed in the Appendix due to space constraints. We believe that these results do not add anything substantial to the discussion, since all models perform similarly on this dataset. Moreover, FLChain is, in our view, too low-dimensional to justify the use of neural networks.
> >
> > > Methods and datasets are not uniform across the paper e.g. table 2, 3 and 4 are of different subsets. If space is limited, is it possible to provide complete table of results comparison of all metrics, all methods and all datasets in the Appendix.
> >
> > Even though a complete table of results might permit a faster comparison across all experiments, providing such a table is impossible due to (i) the lack of clustering labels for real-world datasets and, most importantly, (ii) space constraints. In particular, none of the clustering metrics could be evaluated on the real-world data, since there are no ground truth cluster labels to evaluate against. Hence, merging Table 3 with Table 4 would not be feasible. Additionally, since DSM and SCA had similar performance on (semi-)synthetic data, we do not deem it necessary to adapt SCA’s code (written in Tensorflow 1.8, not compatible with our codebase) to convolutional architectures and augmentations for the comparison on NSCLC CT data. As a final note, the complete table would contain too many columns and rows to be fitted into the width of any page (either main paper or Appendix). Hence, we believe that the results, as they are presented currently, allow making the paper more concise and its message clearer.
> >
> >
> > > Sometimes the results lack immediate clear interpretation in the main text and referring to Appendix seems to be too frequent. It will be better if the main content could be more targeted and focused.
> >
> > We agree that referring often to the Appendix might distract the reader. We would like to stress, however,  that the paper is entirely readable without consulting the Appendices and that the references are included solely for the interested reader desiring more context. All results reported in the Appendices are additional and are not required to follow the discussion and conclusions. We also wanted to make sure that the paper is reproducible and the reader who wants to replicate our experiments could easily find details from the main text.
> >
> > > Would you pls comment how the performance of different models vary by the %censored of the datasets?
> >
> > As can be seen in Table 2, the experiments were performed on the datasets with varying fractions of censored data points, ranging from 25 to 90%. In terms of time-to-event prediction, all methods seem to be able to adequately handle censoring. On Hemodialysis, which is by far the most heavily censored, SCA and DSM are considerably worse calibrated than Weibull AFT or VaDeSC. For DSM, we observed that discounting could mitigate miscalibration to an extent. Since other models do not discount observations, we did not use this trick for the sake of fair comparison. To sum up, we did not observe any particularly alarming or unexpected behaviours.
> >
> > In addition, during the discussion period, we have performed a controlled experiment to study the behaviour of VaDeSC under different percentages of censoring (10-90%) on the synthetic dataset. We have amended these results to the manuscript in Appendix H.4. For reference, we have evaluated Weibull AFT and DSM models as well. In short, VaDeSC stably achieves 90% clustering accuracy for the censoring levels of 10-50%. For > 50% censored observations, its accuracy drops considerably; however, on average, VaDeSC is still slightly better and less variable than the completely unsupervised VaDE (even for 90% of censored observations). Regarding time-to-event prediction, all three models scale and behave very comparably and in an expected manner: with an increase in the percentage of censored data points, CI decreases, RAE on non-censored data increases, and RAE on censored data decreases. These conclusions are similar to those made from the real-world data in the main paper.

---

### Official Review · Reviewer_6hmP · 2021-11-04

**Correctness:** 4
**Technical Novelty And Significance:** 3
**Empirical Novelty And Significance:** 4
**Recommendation:** 8
**Confidence:** 4

**Main Review:**

his paper proposes a new latent variable model for clustering survival data. The new method aims to identify clusters that have characteristic generative survival mechanisms or associations between the auxiliary covariates and the survival times, contrary to methods that identify clusters solely based on the survival times or solely based on covariates. The model is formulated as a latent variable model that combines a general latent variable model with neural network parameterized decoder likelihood for covariates as well as Weibull distribution model for the survival function. Model inference is developed suing the auto-encoding variational Bayes. Amortised variational inference combines a neural network parameterized encoder for the covariates as well as a separate encoder block for the cluster assignments that is defined directly by the generative model, which also naturally handles data points where the survival time is missing/censored. Overall, the model specification seems well-motivated, and the description of the model/results admirably clear.

The proposed method is compared against a large collection of previous methods on both synthetic and real data sets. Performance metrics include clustering assigngment accuracy and survival time prediction accuracy. Overall, experiments with simulated and real data demonstrate that the proposed method can provide more accurate/meaningful cluster assignments than previous methods, while at the same time providing survival predictions that are on par with the existing methods.

This manuscript proposes an interesting latent mixture survival model that makes a useful contribution to the field. Technical details appear correct.  I do not have any major concerns with the manuscript.

Concerning the experiments with simulated data, the differences in cluster-specific Kaplan-Meyer curves appear perhaps unreasonably large compared to what is typically expected in real applications.  Can you provide additional experiments with modified versions of the data set where differences between the KM curves would be smaller?  Results for such simulations would also reveal how the performance of different methods behave when you make your dataset more difficult.


**Summary Of The Paper:**

This paper proposes a new latent variable model for clustering survival data. Experiments with simulated and real data demonstrate that the proposed method can identify accurate/meaningful cluster assignments while simultaneously providing survival predictions that are on par with the existing methods.

**Summary Of The Review:**

This manuscript proposes an interesting latent mixture survival model that makes a useful contribution to the field.

---

> ### Author Response · Authors · 2021-11-13
> **Response to Reviewer 6hmP**
>
> We thank the reviewer for the feedback and thorough review. We made several revisions in the manuscript, which are briefly summarised in the official comment to all of the reviewers. Below is our point-by-point response to your concern.
>
> > Concerning the experiments with simulated data, the differences in cluster-specific Kaplan-Meyer curves appear perhaps unreasonably large compared to what is typically expected in real applications.
>
> We agree that in real-world clinical applications the clusters could be characterised by less disparate KM curves (see Figure 1). Indeed, even in the survMNIST experiment some of the ground truth clusters show very similar KM curves (see pink and brown curves of Figure 4 in the revised manuscript). VaDeSC successfully differentiates all clusters, even those characterised by similar survival distributions.
> Additionally, we tested out the suggested scenario in more detail using the synthetic dataset (see Appendix E.1 and Figure 9), ​​where the ground truth clusters are characterised by survival distributions with less different KM curves, yet disparate generative mechanisms. The quantitative results are shown in Table 3, while the qualitative results can be found in Figure 12, Appendix H.2. Observe that the baselines tend to discover clusters with disparate curves. On the contrary, the proposed approach overcomes this limitation by maximising the joint likelihood of covariates and survival time. As a result, VaDeSC identifies the true clusters, which are characterised by different associations between the covariates and survival time, even though they have rather similar KM curves.

---

### Official Review · Reviewer_k5xJ · 2021-11-07

**Correctness:** 4
**Technical Novelty And Significance:** 3
**Empirical Novelty And Significance:** 3
**Recommendation:** 8
**Confidence:** 2

**Main Review:**

The novelty of the proposed work is modest. As the authors point out, similar clustering-based approaches for survival analysis have already been proposed. Likewise, encoding data types such as images into a latent representation using neural networks is common. To the best of my knowledge, though, combining these approaches is novel.

In terms of technical soundness, the proposed approach is reasonable. I didn’t verify it in detail, but the ELBO approximation appears to follow typical conventions. The experiments are well designed with good baselines and evaluation methodology. The results seem to be presented fairly, and the authors discuss both the strengths and limitation of their approach.

The paper is generally well written and easy to follow.

In terms of reproducibility and contribution to the community, the authors provide technical details in the appendix. They also make their code available. While I did not run the code, it does adhere to a clear style, so I believe that would be a helpful contribution to the community.

**Summary Of The Paper:**

In this work, the authors propose a model for clustering survival data which accounts for both survival time as well as covariates, such as patient demographics. The model is similar to previous approaches, though it incorporates a neural network encoder to handle unstructured data types as input. A modest set of experiments suggest that the proposed approach identifies more meaningful clusters than similar approaches. Also, while not the focus of the proposed approach, empirical evaluation shows that the proposed method performs similarly to existing approaches on time-to-event predictions.

**Summary Of The Review:**

Overall, the proposed work is an incremental step forward for survival analysis. While the impact may not be huge, I believe the proposed approach is a reasonable, novel view on the problem. The experimental results are not overwhelming, but they show that the proposed approach does succeed at clustering patients. This is key for precision medicine, so other researchers could also build on this formulation and approach.

---

> ### Author Response · Authors · 2021-11-13
> **Response to Reviewer k5xJ**
>
> We thank the reviewer for the positive feedback. We agree that an important contribution of our work is a novel perspective on the problem of clustering survival data and hope that follow-up work, particularly by practitioners, could successfully leverage the proposed formulation and method. We made several revisions in the manuscript which are briefly summarised in the official comment to all of the reviewers.

---

> > ### Comment · Reviewer_k5xJ · 2021-11-22
> > **No change after feedback**
> >
> > Hi,
> >
> > Thanks for summarizing the major changes in the updated version. After reading that, as well as the other reviews and author feedback, I still recommend to accept the paper.

---

### Author Response · Authors · 2021-11-13
**To All Reviewers: Summary of Revisions**

Dear reviewers,

We would like to thank all of you for the constructive feedback. Below is a summary of revisions addressing some of your concerns:

- We have extended the discussion of the related work and stressed the differences between VaDeSC and the concurrent work on DCMs by Nagpal et al. (2021) in Section 2.

- We have provided a detailed discussion highlighting the differences between VaDeSC and related techniques, examining generative process assumptions. The discussion can be found in Appendix C.

- We have updated Figure 4 to make it clearer and more interpretable.

- We have performed additional controlled experiments on the clustering and predictive performance of VaDeSC under different censoring percentages. The results are reported in Appendix H.4.

- We have investigated the stability of VaDeSC and DSM on NSCLC CT data. The results are reported in Appendix H.10.

Please see our point-by-point responses to each reviewer for a more detailed discussion.

References:

Nagpal, C., Yadlowsky, S., Rostamzadeh, N. and Heller, K., 2021. Deep Cox mixtures for survival regression.

---

### Decision · Program_Chairs · 2022-01-20

**Decision:**

Accept (Poster)

**Comment:**

Four knowledgeable referees recommend Accept. I also think the paper provides a unique contribution to the field of deep survival models and I, therefore, recommend Accept